# Integrating genetic regulation and single-cell expression with GWAS prioritizes causal genes and cell types for glaucoma

Andrew R. Hamel [1,2,3], Wenjun Yan [4,62], John M. Rouhana[1,2,3,62], Aboozar Monovarfeshani[4,62], Xinyi Jiang [5,6], Puja A. Mehta [1,2,3], Jayshree Advani[7], Yuyang Luo[1,2,3], Qingnan Liang [8], Skanda Rajasundaram[2,9,10], Arushi Shrivastava [1,2,3], Katherine Duchinski [1,2,3,11], Sreekar Mantena [1,12], Jiali Wang [1,2,3], Tavé van Zyl[4,13], Louis R. Pasquale[14], Anand Swaroop [7], Puya Gharahkhani [15], Anthony P. Khawaja [16], Stuart MacGregor [15], International Glaucoma Genetics Consortium (IGGC)*, Rui Chen [8], Veronique Vitart [5], Joshua R. Sanes [4], Janey L. Wiggs [1,2,3] & Ayellet V. Segrè [1,2,3] ✉

Primary open-angle glaucoma (POAG), characterized by retinal ganglion cell death, is a leading cause of irreversible blindness worldwide. However, its molecular and cellular causes are not well understood. Elevated intraocular pressure (IOP) is a major risk factor, but many patients have normal IOP. Colocalization and Mendelian randomization analysis of >240 POAG and IOP genome-wide association study (GWAS) loci and overlapping expression and splicing quantitative trait loci (e/sQTLs) in 49 GTEx tissues and retina prioritizes causal genes for 60% of loci. These genes are enriched in pathways implicated in extracellular matrix organization, cell adhesion, and vascular development. Analysis of single-nucleus RNA-seq of glaucoma-relevant eye tissues reveals that the POAG and IOP colocalizing genes and genome-wide associations are enriched in specific cell types in the aqueous outflow pathways, retina, optic nerve head, peripapillary sclera, and choroid. This study nominates IOP-dependent and independent regulatory mechanisms, genes, and cell types that may contribute to POAG pathogenesis.

Primary open-angle glaucoma (POAG) is the leading cause of irreversible blindness worldwide among people over the age of 55[1]. It is characterized by progressive optic neuropathy, caused by the gradual death of retinal ganglion cells (RGCs) that transmit visual information from the outer retina to the brain via the optic nerve (myelinated RGC axons)[2]. Elevated intraocular pressure (IOP) is a major risk factor for POAG[3] and is primarily caused by decreased outflow of the aqueous humor from the ocular anterior segment. Decreased outflow may be due to abnormal function of structures in the anterior segment of the eye, consisting of the trabecular meshwork (TM)[4] and Schlemm's canal

(SC)[5] in the conventional outflow pathway, and the ciliary muscle and iris in the uveoscleral (unconventional) pathway[6]. However, about one third of patients with POAG display optic nerve degeneration in the absence of abnormally high IOP measurements (normal tension glaucoma (NTG))[7]. Conversely, many people with elevated IOP do not develop glaucoma, suggesting that other processes, including increased RGC susceptibility to normal IOP, might also lead to optic nerve damage. Currently, neuroprotective therapies are lacking, and medications that reduce IOP have limited effectiveness[2]. Gaining a better understanding of the molecular and cellular causes of POAG in

A full list of affiliations appears at the end of the paper. *A list of authors and their affiliations appears at the end of the paper.
✉e-mail: ayellet_segre@meei.harvard.edu

the anterior and posterior segments of the eye could suggest novel therapeutic targets.

A recent multi-ethnic genome-wide association study (GWAS) meta-analysis of 34,179 POAG cases and 349,321 controls of European, Asian, and African ancestries identified 127 risk loci associated with POAG[8], explaining ~9% of POAG heritability, and a meta-analysis of the European subset identified 68 POAG loci[8], some of which were not uncovered in the cross-ancestry meta-analysis. Furthermore, a GWAS meta-analysis of IOP performed on 139,555 individuals, primarily of European descendant[9], has identified 133 independent associations in 112 loci, largely overlapping with two other studies[10,11]. The IOP variants' effect sizes and direction of effect are highly correlated with their effect on POAG risk[8,9], and together they explain 9-17% of IOP heritability. Vertical-cup-to-disc ratio (VCDR), central corneal thickness, and corneal hysteresis, a measure of the viscoelastic damping of the cornea, have also been associated with POAG risk, and large GWAS meta-analyses have uncovered 70-200 genetic associations for these traits[12–23].

Identifying putative causal genes and cell types underlying the genetic associations with POAG and its related traits is challenging. As with other complex traits, a majority of associated variants lie in noncoding regions and are enriched for regulatory effects[24–26]. Due to linkage disequilibrium (LD), the discovered associations typically tag multiple variants and genes, making it hard to pinpoint the implicated causal gene(s) from sequence alone. Furthermore, genetic regulatory effects in relevant ocular tissues are limited, reported to date only in retinal tissues[27–30], and have not yet been detected at cellular resolution in other parts of the eye. Nevertheless, through single-cell or single-nucleus RNA-sequencing (sc/snRNA-seq), human cell atlases and cellular level transcriptomes have been generated for various non-diseased eye tissues relevant to POAG pathogenesis, including retina[31–33], the aqueous humor outflow pathways[34,35], six tissues in the anterior segment[36], and the optic nerve head (ONH), where RGCs pass to exit the eye, the optic nerve, and surrounding posterior tissues[37]. Using a method we recently developed, ECLIPSER (Enrichment of Causal Loci and Identification of Pathogenic cells in Single Cell Expression and Regulation data)[38,39], we show that cell type-specific enrichment of genes mapped to GWAS loci of complex diseases and traits can help identify cell types of action for diseases in relevant tissues[38,39].

In this study, we combine expression quantitative trait loci (eQTLs) and splicing QTLs (sQTLs) in 49 (non-ocular) tissues from the Genotype-Tissue Expression (GTEx) Project[26], retinal eQTLs[27], retinal Hi-C data[40], and single-nucleus expression from glaucoma-relevant eye tissues[33,36,37] with POAG and IOP genetic associations to identify regulatory mechanisms, genes, pathways, and cell types that may play an important role in POAG etiology. Using two Bayesian-based colocalization methods, eCAVIAR[41] and *enloc*[42], followed by two-sample Mendelian Randomization[43], we identify putative causal genes and direction of regulatory effects on POAG and IOP for over half the GWAS loci. These genes are enriched in previously implicated and new biological processes. By testing for cell type-specific enrichment of POAG and IOP colocalizing genes in the ocular anterior segment, retina, and optic nerve head (ONH) and surrounding posterior tissues, using ECLIPSER, we identify known and less well-established pathogenic cell types for POAG, including fibroblasts in the conventional and unconventional outflow pathways and in the peripapillary sclera encompassing the ONH, and macroglial cells in the retina and ONH, suggesting both IOP-dependent and neuroprotective processes. These results are supported by two additional cell type enrichment methods that consider genome-wide associations with POAG and IOP, S-LDSC[25] and MAGMA[44].

## Results

An overview of the analytical steps and approaches taken are described in Fig. 1 and Supplementary Note 1.

## POAG and IOP associations enriched among eQTLs and sQTLs

To assess the relevance of eQTLs and sQTLs to POAG risk and IOP variation, we tested whether *cis*-eQTLs and *cis*-sQTLs (e/sQTLs) from 49 GTEx (v8) tissues[26] and peripheral retina *cis*-eQTLs[27] were enriched for POAG or IOP associations (GWAS $P < 0.05$) using *QTLEnrich*[24,26] that adjusts for confounding factors and tissue sample size (Methods, Supplementary Note 2 and Fig. 1a). We found significant enrichment of multiple POAG and IOP associations (both genome-wide significant and subthreshold) among eQTLs and sQTLs in most of the 49 GTEx tissues and in retina (Bonferroni-corrected $P < 5 \times 10^{-4}$) (Fig. 2a, b, Supplementary Figs. 1 and 2, and Supplementary Data 1–3). Many of the top enriched GTEx tissues contain cell types that may be pathogenic to glaucoma (Supplementary Note 2). The relative contribution of sQTLs to POAG and IOP, as measured by adjusted fold-enrichment and estimated true positive rate, was larger than the relative contribution of eQTLs to these traits (One-sided Wilcoxon rank sum test $P < 1.5 \times 10^{-11}$ and $P < 0.03$, respectively; Supplementary Fig. 1a–c and Supplementary Data 1–3), as observed with other complex traits[26]. The absolute number of eQTLs proposed to contribute to POAG and IOP (average 258 to 606 per tissue) was 2-fold larger than that of sQTLs (average 124 to 320 per tissue), likely due to the larger discovery rate of eQTLs compared to sQTLs[26] (Supplementary Fig. 1a–c and Supplementary Data 1–3). The target genes of eQTLs or sQTLs with top-ranked POAG or IOP GWAS *p*-values ($P < 0.05$) were enriched in metabolic and cellular processes (Methods, Supplementary Data 4–5, and Supplementary Note 3).

## Colocalization analysis of POAG and IOP GWAS loci with *cis*-e/sQTLs

Given the widespread e/sQTL enrichment of POAG and IOP associations, we used the e/sQTLs in all 49 GTEx tissues and retina eQTLs to propose putative causal genes that may underlie genome-wide significant loci for these traits. We applied two colocalization methods, eCAVIAR[41] and *enloc*[42], to 127 POAG loci from a large cross-ancestry GWAS meta-analysis[8], 68 POAG loci from a European (EUR) subset meta-analysis[8] (POAG EUR), and 133 IOP loci from a primarily European GWAS meta-analysis[9] (IOP) (variant list in Supplementary Data 6; Methods and Fig. 1b), and any e/sQTLs that overlapped each GWAS locus LD interval (Methods). The results are presented per trait and colocalization method in Supplementary Data 7–12 and summarized in Supplementary Data 13. We defined a "comprehensive set" of putative causal genes and regulatory mechanisms for POAG and IOP as those e/sGenes that were significant with at least one of the colocalization methods (Colocalization posterior probability (CLPP) > 0.01 for eCAVIAR, and regional colocalization probability (RCP) > 0.1 for *enloc* (see Methods); Supplementary Data 13), filtering out potential false positives (See Methods and examples in Supplementary Fig. 3). The largest number of colocalizing e/sGenes was found in tibial nerve, adipose, skin, artery, and fibroblasts, among other tissues (Supplementary Fig. 4), many of which contain cell types relevant to the pathogenicity of glaucoma. Eighteen retina eQTLs colocalized with 13 POAG and/or IOP loci (Column AH in Supplementary Data 13). The number of significantly colocalizing e/sGenes per tissue significantly correlated with tissue sample size (Pearson's $R^2 = 0.72$, $P = 1 \times 10^{-14}$, Supplementary Fig. 4) that is also associated with the number of detected e/sQTLs per tissue[26]. This suggests that e/sQTL discovery power is a driving factor in tissue identity of the colocalizing e/sQTLs. We therefore primarily considered the causal genes proposed by the colocalization analysis and not the associated tissues, in downstream analyses.

We found that 58% of all GWAS loci tested significantly colocalized with at least one eQTL and/or sQTL based on eCAVIAR and/or *enloc*: 60% (76) of 127 cross-ancestry POAG GWAS loci, 53% (36) of 68 European POAG loci and 59% (79) of 133 IOP loci (Fig. 2c and Supplementary Data 14). About 55% and 29% of GWAS loci colocalized with ≥1 eQTL and ≥1 sQTL, respectively. For 21% of the POAG and IOP GWAS

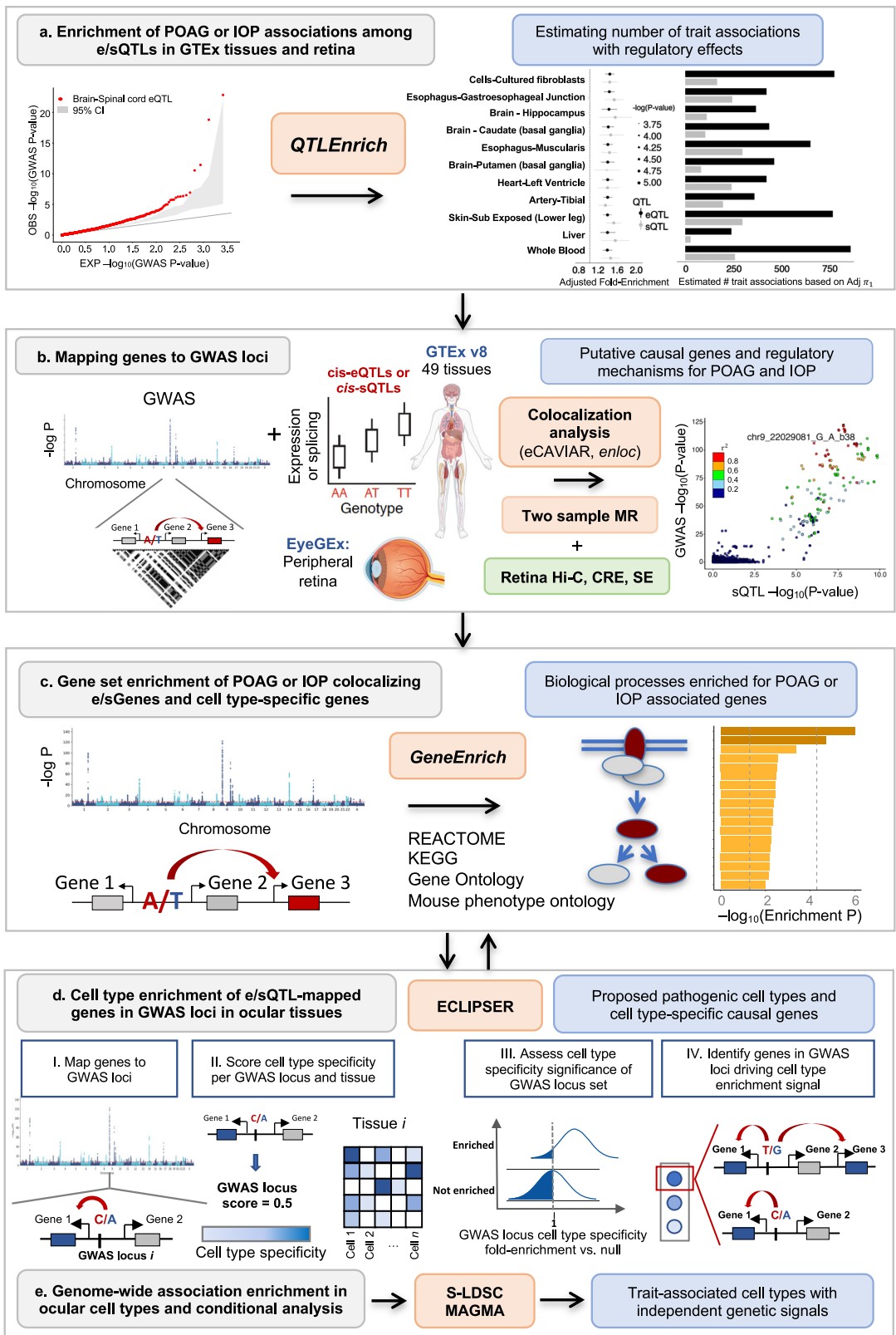

loci (69 loci total), significant colocalization was found for the same e/sGene with eCAVIAR and *enloc* ('high confidence set' listed in Table 1 and Supplementary Data 15). The GWAS-e/sQTL colocalization analysis significantly reduced the number of putative causal genes per GWAS locus for POAG and IOP from an average of 22.8 ± 1.8 genes tested per LD interval (range: 3-166, median = 15) to an average of 3.5 ± 0.4 genes per locus (range: 1–36, median of 1 or 2 genes per locus per trait;

Fig. 2d, e and Supplementary Data 16). eQTLs and sQTLs nominated an average of 3 and 2 causal genes per locus, respectively, with partial overlap of target genes between the colocalizing eQTLs and sQTLs (Fig. 2e and Supplementary Data 16). 60-72% of the colocalizing e/sGenes per trait were protein-coding and 18-20% were noncoding RNA genes, half of which were lincRNAs and half antisense genes (Fig. 2f, Supplementary Data 17, 18, and Supplementary Note 4). A single causal

**Fig. 1 | Analysis workflow from POAG and IOP GWAS to causal regulatory mechanisms, genes, pathways, and cell types. a** POAG and IOP associations genome-wide (known and modest associations) were tested for enrichment among expression and splicing quantitative trait loci (e/sQTLs) in GTEx tissues and retina compared to permuted null sets of variants matched on confounding factors, using *QTLEnrich* (one-sided). In cases where enrichment was found, the lower bound number of e/sQTLs in a given tissue, likely to be true trait associations was estimated using an empirically derived, true positive rate ($(\pi_1)$) approach. **b** Putative causal genes were prioritized per known POAG and IOP genome-wide association study (GWAS) locus by applying two colocalization methods (eCAVIAR, enloc) to all e/sQTLs from 49 GTEx tissues and retina eQTLs that overlapped each locus, followed by two-sample Mendelian Randomization (MR). Overlap of the colocalizing GWAS loci and e/sQTLs with Hi-C (3D chromosome conformation capture), *cis*-regulatory element (CRE), and super-enhancer (SE) regions from human retina was utilized to further prioritize causal genes. The human and eye images were created with BioRender.com. **c** All target genes of significantly colocalizing e/sQTLs (e/

sGenes) or cell type-specific genes per trait were tested for enrichment in signaling and metabolic pathways (Reactome, KEGG), gene ontologies, and mouse phenotype ontologies using *GeneEnrich* (one-sided). The POAG cross-ancestry GWAS meta-analysis Manhattan plot was generated using QMplot (https://github.com/ShujiaHuang/qmplot). **d** Significantly colocalizing e/sGenes were tested for enrichment in specific cell types in single-nucleus RNA-seq data of glaucoma-relevant eye tissues, using ECLIPSER (one-sided). Cell type-specific genes were defined with cell type fold-change>1.3 and FDR < 0.1 per tissue. Cell type-specificity significance per GWAS locus set for a given trait was assessed against a null distribution of loci associated with unrelated, non-ocular traits, using a Bayesian Fisher's exact test. Genes mapped to GWAS loci with a cell type-specificity score above the 95th-percentile of null locus scores were proposed as contributing to the trait in the enriched cell type. **e** Cell type enrichment for the POAG and IOP GWAS was corroborated using two regression-based methods that assess cell type-specificity of trait associations considering all associations genome-wide: stratified-LD score regression (S-LDSC) and MAGMA.

gene was proposed for 80 (42%) of the POAG and IOP loci with significantly colocalizing e/sQTLs (Supplementary Data 19, 20), 49 (61%) of which are the nearest gene to the lead GWAS variant. In 30.3% (23/76), 16.7% (6/36) and 31.7% (25/79) of the POAG cross-ancestry, POAG EUR and IOP GWAS loci, respectively, with significant colocalization results, the colocalizing e/sGenes were not the nearest gene to the lead GWAS variant. In total, 228, 118, and 279 genes, including previously suggested and novel ones, are candidate causal genes for POAG cross-ancestry risk, POAG EUR risk, and IOP variation, respectively (Supplementary Data 21), with a total of 459 genes proposed from the combined datasets.

## Colocalizing e/sGenes of top POAG and IOP GWAS loci and direction of regulatory effect on disease risk

In addition to prioritizing causal genes and regulatory mechanisms that may contribute to POAG and IOP, colocalizing e/sQTLs propose the direction of effect of altered gene expression or splicing on disease risk or trait variation (examples for top POAG and IOP GWAS signals in Fig. 3 and Supplementary Fig. 5). For example, an eQTL and sQTL acting on *TMCO1* and an eQTL acting on *TMCO1*'s antisense, *RP11-466F5.8*, in the opposite direction, colocalized with the second strongest association with POAG (lead variant rs2790053, odds ratio (OR) = 1.35. CLPP = 0.92-1) and the top IOP associations (two LD-independent variants: rs116089225, beta = −0.744 and rs10918274, beta = 0.377; CLPP = 0.87−1) (Fig. 4a–e and Supplementary Fig. 6). Decreased expression of *TMCO1* and increased expression of *RP11-466F5.8* are proposed to lead to increased IOP levels and increased POAG risk (Fig. 4, Supplementary Fig. 6, and Supplementary Data 13). Furthermore, an alternative splice donor site in exon 4, the first exon in the *TMCO1* mRNA in GTEx Cells-Cultured fibroblasts, leads to a longer exon 4 (Fig. 4f,g) that is associated with decreased POAG risk (Supplementary Data 7 and 13). *TMCO1* is expressed in different cell types in the anterior and posterior parts of the eye, including lymphatic and fibroblast cells in the conventional and unconventional outflow pathways, vascular and immune cells in the anterior and posterior segments, and macroglial cells in the retina (Supplementary Fig. 7). Other examples include *ANGPT1* and *ANGPT2*, involved in vascular biology, whose increased expression is proposed to reduce IOP levels (Supplementary Fig. 8 and Supplementary Data 13) that is consistent with the effect observed on IOP in *Angpt1*-knockout mice[45].

## Colocalizing genes for shared and distinct POAG and IOP loci

Of 50 overlapping POAG and IOP GWAS loci, 39 (78%) of the loci had at least one significant colocalization result for both traits, and in all cases at least one common gene was implicated (Supplementary Data 13 and 15, and Table 1). In most of the cases (95%), the relative direction of effect of the colocalizing e/sQTLs on IOP was consistent with IOP's effect on POAG risk, proposing IOP-dependent mechanisms for POAG

risk. For example, decreased *GAS7* expression or increased *ABO* expression were associated with both increased IOP levels and increased POAG risk (Supplementary Data 7–13 and Supplementary Figs. 9, 10). e/sGenes that colocalize with POAG loci not associated with IOP (48 loci; Column N in Supplementary Data 13) may suggest IOP-independent mechanisms.

To prioritize regulatory variants and genes that may affect POAG independent of IOP, we integrated retina Hi-C loops and/or epigenetically derived *cis*-regulatory elements (CREs) and super-enhancers (SEs) (Supplemental Data 4 in Marchal et al.[40]) with POAG-only loci (Methods). In 17 of the loci, ≥1 colocalizing e/sQTLs was supported as a potential causal mechanism by retina Hi-C loops (6 loci), CREs (16 loci) and/or SEs (5 loci) (Supplementary Data 22). This includes the strongest normal tension glaucoma (NTG) association (9p21)[8,46] in the POAG cross-ancestry (rs944801 OR = 1.26) and POAG EUR (rs6475604 OR = 1.3) GWAS meta-analyses that colocalized with a *CDKN2A* eQTL in brain cortex, *CDKN2B-AS1 and RP11-149I2.4* sQTLs in pituitary, and *CDKN2B* eQTL in skeletal muscle (Supplementary Data 13). The POAG risk variants and colocalizing e/sQTLs in this locus overlapped retinal CREs (Fig. 5a). The e/sQTL results imply that increased expression of *CDKN2A*, decreased expression of *CDKN2B*, and exon skipping in *CDKN2B-AS1* may increase POAG risk (Fig. 3, Supplementary Figs. 5a and 11). Other examples, involving a retina *SLC2A12* eQTL overlapping a retinal CRE, and e/sQTLs acting on *RERE* and its antisense, *RERE-AS1*, that are physically linked via retina chromatin loops to the *RERE* transcription start site (TSS), are shown in Fig. 5b and 5c, respectively (see also Supplementary Data 22). Notably, *RERE* expression is enriched in oligodendrocytes in the optic nerve head and optic nerve (False discovery rate (FDR) = 0.07) and in retinal pigment epithelium (RPE) and S-cones in the macula (FDR = 0.06), as shown below (Supplementary Data 35; Supplementary Fig. 18f, g).

## Colocalizing genes in population-specific and cross-ancestry POAG loci

For all 59 POAG EUR loci also found in the POAG cross-ancestry meta-analysis, at least one common colocalizing e/sGene was found per locus for both the EUR and cross-ancestry GWAS (Supplementary Data 13). One such example is *EFEMP1*, in which rare mutations have been associated with a Mendelian form of glaucoma[47]. Colocalization analysis suggests that skipping of exons 6 and 7 in *EFEMP1* may be protective for POAG (Supplementary Fig. 12). Of the 9 loci found only in the POAG EUR GWAS, two loci colocalized with eQTLs acting on several genes each, including genes involved in the extracellular matrix (*EMID1*) and vascular endothelial growth (*ANGPTL2*), respectively, both of which also colocalized with IOP (Supplementary Data 23). In addition, three associations in the POAG cross-ancestry meta-analysis demonstrated significant allelic heterogeneity among the three populations (European, East Asian, and African American)[8]. e/sQTL

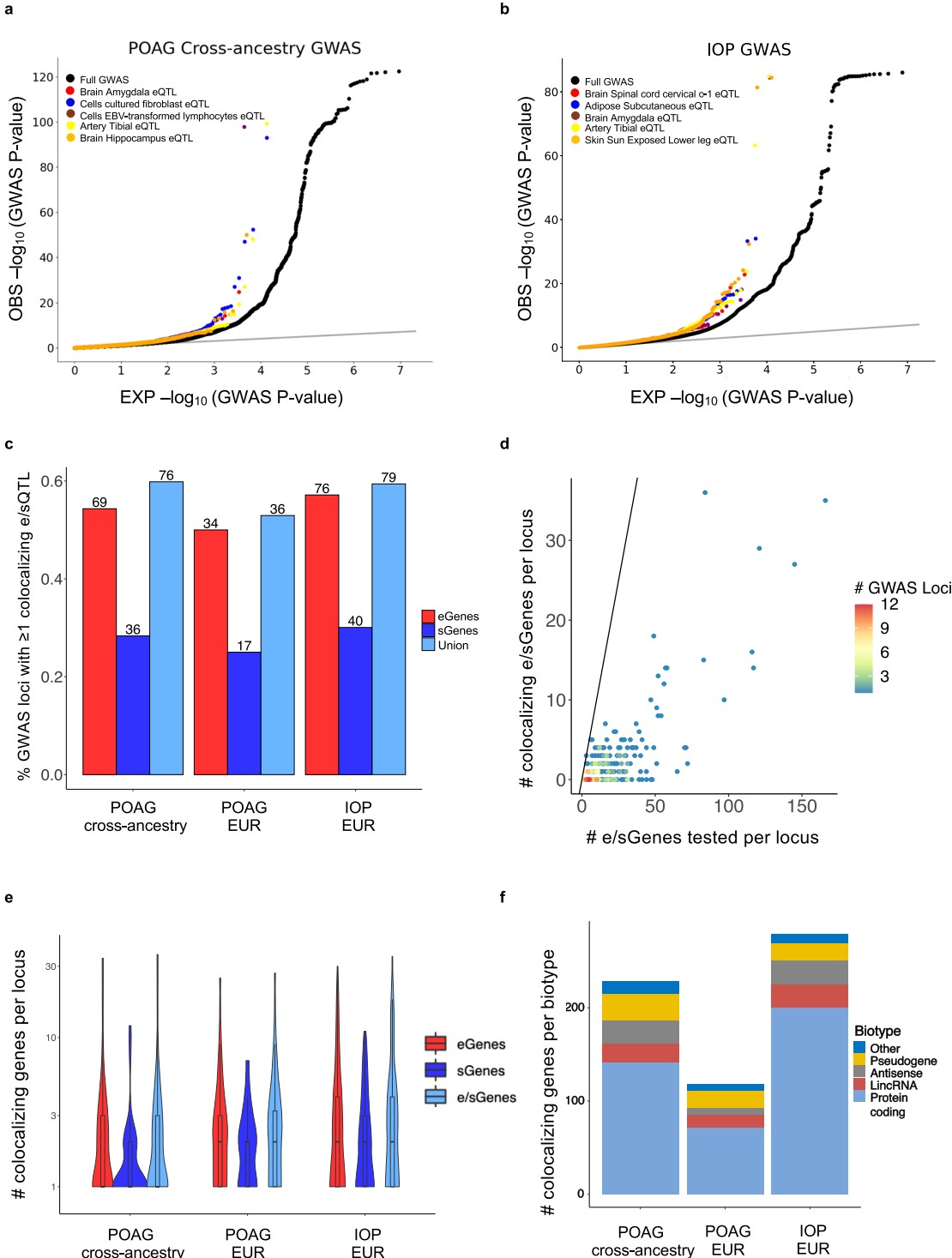

colocalized with two of these loci. One is the 9p21 locus rs944801 with *CDKN2A/B* described above, which was significant in the European and Asian populations, but not the African population (P_heterogeneity = $1.5 \times 10^{-8}$) in which it is in lower frequency (AFR MAF = 0.073, EUR MAF = 0.42, EAS MAF = 0.11 in gnomAD, https://gnomad. broadinstitute.org/). The other is a European-specific locus with the largest POAG odds ratio (rs74315329, OR = 5.47). This variant is a nonsense mutation (p.Gln368Ter) in *MYOC*, known to cause juvenile-onset and adult-onset open-angle glaucoma with dominant inheritance[48]. This variant is 10-fold more common in the European

population compared to the African population and is not found in the East Asian population (gnomAD). Of all the e/sQTL gene-tissue pairs that overlapped this locus targeting 24 genes, we identified an sQTL acting on *PIGC*, phosphatidylinositol glycan anchor biosynthesis class C, that significantly colocalized with the POAG cross-ancestry locus (spleen CLPP = 0.12, and arterial tissues RCP = 0.26-0.34; Supplementary Data 13 and Supplementary Fig. 13). Conditioning on the nonsense variant in *MYOC* that is likely the primary causal variant in the locus, we found that a secondary haplotype colocalized with the *PIGC* sQTL (Supplementary Data 24–28; more details in Supplementary Note 5).

**Fig. 2 | Enrichment and colocalization analysis of eQTLs and sQTLs with POAG and IOP associations.** Quantile-quantile (Q-Q) plots of POAG cross-ancestry (**a**) and IOP (**b**) GWAS -$\log_{10}$ (*P*-value) compared to expectation for the best eQTL per eGene sets (eVariants with FDR < 0.05) of the most significantly enriched tissues based on adjusted fold-enrichment (colored points), compared to all variants in the GWAS (black points). Grey line represents the diagonal. *QTLEnrich* (one-sided test; see Methods) was applied to assess GWAS *p*-value enrichment among the eQTL sets. Bonferroni correction was applied. **c** Histogram of percent of GWAS loci with ≥1 colocalizing e/sQTL (colocalization posterior probability (CLPP) > 0.01 from eCAVIAR and/or regional colocalization probability (RCP) > 0.1 from *enloc*) for POAG cross-ancestry, POAG European (EUR) ancestry subset, and IOP European ancestry GWAS meta-analyses. Numbers above the bars represent the number of loci with at least one colocalizing e/sQTL. Red, dark blue, and light blue bars indicate percentage of loci with at least one colocalizing eGene, sGene, or both, respectively. e/sGene, gene with at least one significant e/sQTL. **d** Scatter plot

comparing unique number of e/sGenes that significantly colocalized per GWAS locus versus unique number of e/sGenes tested per locus. Points are color-coded by number of GWAS loci. The black line represents the diagonal. **e** Violin plots showing the distribution of the unique number of colocalizing eGenes (red), sGenes (dark blue), or both (light blue) per locus for the three GWAS meta-analyses tested. The number of GWAS loci that colocalized with eGenes, sGenes, or both shown in the violin plot are *N* = 69, 36, 76, respectively, for POAG cross-ancestry, *N* = 34, 17, 36, respectively, for POAG EUR, and *N* = 76, 40, 79, respectively, for IOP (Supplementary Data 16). The center line in the box plots contained within each violin plot shows the median, the box edges depict the interquartile range (IQR), and whiskers mark 1.5x the IQR. The violin plot edges represent the minima and maxima values. **f**, Stacked histogram showing the number of colocalizing e/sGenes per gene biotype for each GWAS. Protein coding (light blue), lincRNA (brown), antisense (grey), pseudogenes (yellow), and other (dark blue).

## Table 1 | List of high-confidence colocalizing expression and splicing QTLs with POAG and IOP GWAS loci

| Lead GWAS variant[a] | RS ID | Nearest Gene/s | Total # colocalizing e/sGenes[b] | Colocalizing GTEx e/sQTLs based on eCAVIAR and *enloc* and significant MR[c] | Colocalizing retina eQTL (eCAVIAR)[d] | POAG GWAS ancestry[e] | Colocalizing with IOP GWAS[f] |
|---|---|---|---|---|---|---|---|
| chr1:171636338:G:A | rs74315329 | *MYOC* | 1 | *PIGC (s)** | – | CA | No |
| chr1:165768467:G:C | rs2790053 | *TMCO1, TMCO1-AS1 (RP11-466F5.8)* | 2 | *TMCO1 (e,s), TMCO1-AS1 (e)** | *TMCO1-AS1* | CA,EU | Yes |
| chr9:22051671:G:C | rs944801 | *CDKN2B-AS1* | 3 | *CDKN2B-AS1 (s)* | – | CA,EU | No |
| chr4:7902636:G:A | rs938604 | *AFAP1* | 2 | *AFAP1 (e)** | – | CA | Yes |
| chr17:10127866:G:A | rs9913911 | *GAS7* | 1 | *GAS7 (e)** | – | CA,EU | Yes |
| chr9:126628521:C:T | rs3829849 | *LMX1B* | 4 | *LMX1B (e)* | – | CA,EU | Yes |
| chr7:116522252:T:A | rs10257125 | *CAV1* | 2 | *CAV2 (e,s)** | – | CA,EU | Yes |
| chr11:86657064:C:T | rs10792871 | *ME3* | 4 | *PRSS23 (e)*, ME3 (e)** | *PRSS23* | CA,EU | Yes |
| chr9:133255801:C:T | rs8176749 | *ABO* | 1 | *ABO (e)** | – | CA | Yes |
| chr15:73928957:C:T | rs1550437 | *LOXL1* | 2 | *LOXL1 (e,s)** | – | CA,EU | No |
| chr3:186411027:G:A | rs56233426 | *DGKG* | 2 | *DGKG (e)* | *DGKG* | CA,EU | Yes |
| chr17:46010134:G:A | rs41543317 | *MAPT* | 9 | – | *KANSL1, LRRC37A2, LRRC37A* | EU | No** |
| chr3:188349165:G:T | rs6787621 | *LPP* | 1 | *LPP (e,s)** | – | CA | Yes |
| chr1:36147354:C:A | rs4652902 | *TRAPPC3* | 1 | – | *COL8A2* | CA | Yes |
| chr22:28712241:G:A | rs5752776 | *CHEK2* | 5 | *TTC28 (e)** | – | CA,EU | No** |
| chr1:37605594:C:T | rs6687545 | *GNL2, RSPO1* | 7 | *GNL2 (e,s)*, MEAF6 (e)** | *MEAF6* | EU | Yes |
| chr11:47447887:G:A | rs7111873 | *RAPSN* | 8 | – | *PSMC3* | CA | No** |
| chr2:12811195:C:T | rs12623251 | *TRIB2* | 1 | *TRIB2 (e)* | – | CA,EU | No |
| chr15:57261634:T:A | rs2431023 | *TCF12* | 2 | *ZNF280D (e), TCF12 (e)* | – | CA,EU | No |
| chr11:130412183:C:T | rs2875238 | *ADAMTS8* | 2 | *RP11-121M22.1 (e,s)*, ADAMTS8 (e)* | – | CA | No |
| chr14:74618126:G:A | rs754458 | *LTBP2, AREL1* | 4 | *NPC2 (e)** | *NPC2* | CA | Yes |
| chr7:134835770:C:A | rs10237321 | *CALD1* | 1 | *CALD1 (e)* | – | CA | No |
| chr6:134051012:G:C | rs2811688 | *SLC2A12* | 3 | *SLC2A12 (e)*, TBPL1 (e)* | *SLC2A12* | CA,EU | No |

[a]Variant chromosome positions are in genome build 38.

[b]Total number of target genes of expression and splicing QTLs (e/sGenes) in any of the 49 GTEx tissues or retina that colocalized with a given POAG cross-ancestry GWAS locus based on eCAVIAR (CLPP > 0.01) and/or *enloc* (RCP > 0.1).

[c]e/sGenes from 49 GTEx tissues that significantly colocalized with a POAG cross-ancestry and/or European GWAS locus based on both eCAVIAR (CLPP > 0.01) and *enloc* (RCP > 0.1), and displayed significant Mendelian Randomization (MR) results (FDR < 0.05) using the European POAG GWAS meta-analysis. e, eQTL. s, sQTL.

[d]Target gene/s of retina eQTLs that significantly colocalized (CLPP > 0.01) with POAG cross-ancestry (CA) or European (EU) GWAS loci and displayed significant MR results (FDR < 0.05).

[e]Ancestry of POAG GWAS meta-analysis, cross-ancestry (CA) or European (EU), that colocalized with the e/sQTLs in the given locus. The GWAS variants are ordered based on their POAG effect size (odds ratio) in descending order.

[f]This column states whether the POAG colocalizing e/sQTLs significantly colocalized with the IOP GWAS and showed significant MR results (FDR < 0.05) with IOP. No colocalization with IOP GWAS suggests IOP-independent mechanisms. List of tissues in which the e/sQTLs colocalized with the POAG or IOP GWAS loci is given in Supplementary Data 7-12. *e/sQTLs where pleiotropy-robust sensitivity testing of significant MR results could be performed (≥3 LD-independent e/sVariants (instrumental variables)). All these genes were robust to horizontal pleiotropy (*P* > 0.05). **Loci associated with IOP, but for which colocalization was not found for the specific e/sGenes. A similar list of high-confidence colocalizing e/sQTLs with IOP is given in Supplementary Data 15.

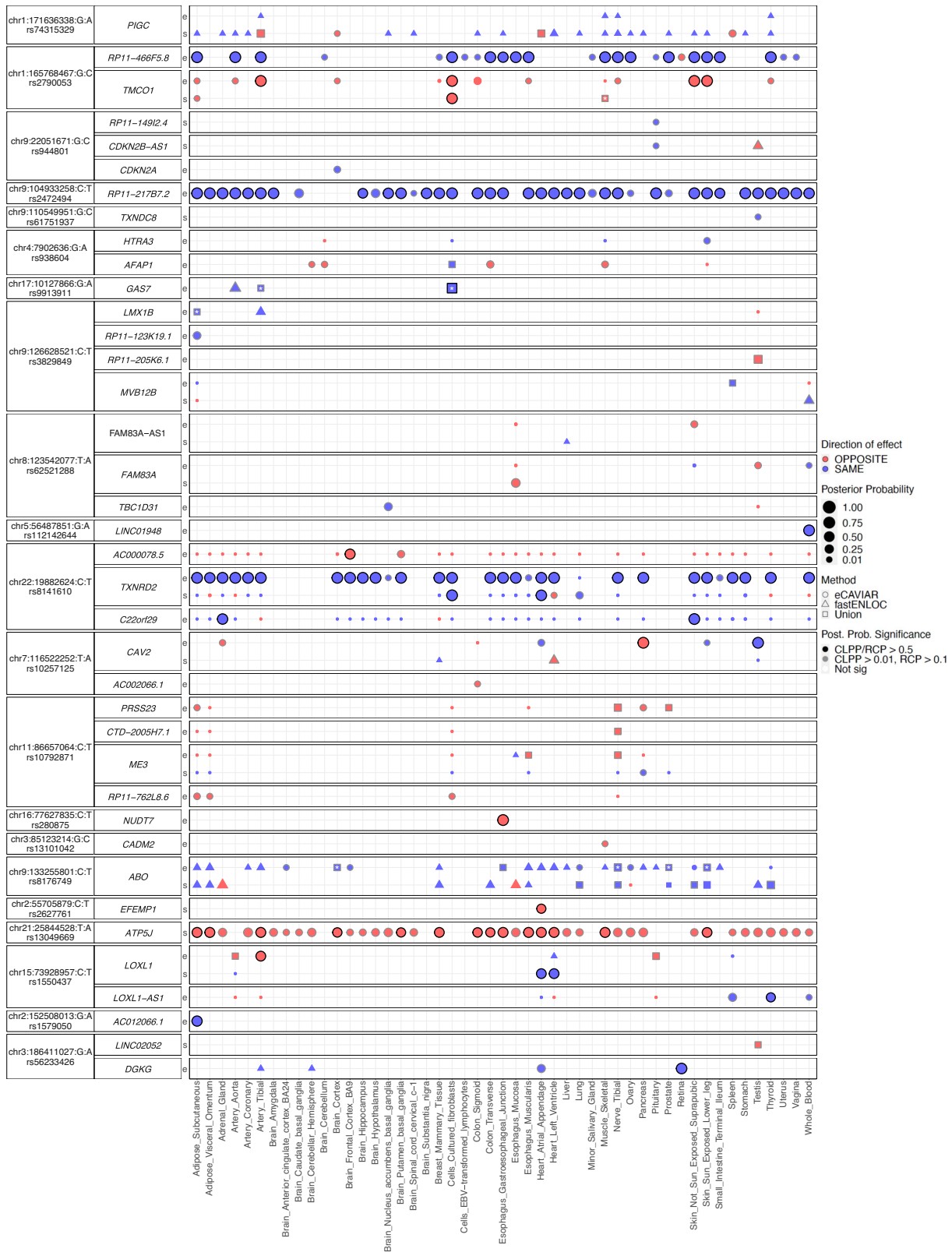

These results suggest that decreased exon 2 skipping in *PIGC* or increased *PIGC* expression may lead to increased POAG risk (Supplementary Fig. 13). *PIGC* is among 10 colocalizing POAG genes enriched in oligodendrocytes in the optic nerve head and optic nerve (See below, FDR = 0.07; Supplementary Data 35 and Supplementary Fig. 18f, g).

## Mendelian randomization (MR) of colocalizing e/sQTLs

To provide additional support for a causal relationship between e/sQTLs and POAG and/or IOP, we applied two-sample MR to all the significantly colocalizing e/sQTL and GWAS locus pairs based on eCAVIAR and/or *enloc*, using the European POAG and IOP GWAS summary statistics (Methods). We found supportive evidence for a

**Fig. 3 | Colocalizing e/sQTLs in GTEx tissues and retina with top POAG GWAS loci.** Genes with at least one significant colocalization result are shown for e/sQTLs tested across 49 GTEx tissues and peripheral retina for the top 21 POAG cross-ancestry GWAS loci. GWAS loci were ordered by absolute value of their effect size. Within each locus, genes were ordered based on their chromosome position. Bubble size is proportional to the maximum colocalization posterior probability of all e/sVariants tested for the given gene, QTL type and tissue combination. Points are color-coded by direction of effect (blue if increased expression or splicing increases POAG risk or vice versa; red if increased expression or splicing decreases

POAG risk or vice versa). Shape of points indicates colocalization method used: circle (eCAVIAR), triangle (enloc), and square (tested in both methods; results shown for method with maximum posterior probability). Grey or black border denotes variant-gene-tissue-QTL combination that passed quality control (QC) filtering (Methods) and a colocalization posterior probability cutoff above 0.01/0.1 (CLPP/RCP) or 0.5 (higher confidence), respectively. White or black asterisk in the square indicates whether the second method tested passed a posterior probability cutoff 0.01/0.1 (CLPP/RCP) or 0.5, respectively.

causal relationship (FDR < 0.05) for 348 (75%) genes that were robust to the influence of horizontal pleiotropy, where pleiotropy-robust sensitivity could be performed (Supplementary Data 29 and Supplementary Discussion). A high-confidence list of putative POAG and/or IOP causal genes based on colocalization analysis and MR is provided in Table 1 and Supplementary Data 15. We found 239 e/sGenes to have significant MR associations with both POAG and IOP, including *TMCO1*, *GAS7*, and *LMX1B*, which colocalized with the largest association signals for both POAG and IOP GWAS loci (Supplementary Figs. 5, 6, 9), and *DGKG* and *NPC2*, whose retina eQTLs colocalized with POAG and IOP. Sixty-eight genes had significant MR associations with IOP but not POAG, such as *HLA-B* and *SLC7A6*, and 41 genes had significant MR associations with POAG but not IOP, such as *CDKN2B-AS1, RERE, and YAP1*, proposing high-confidence IOP-independent mechanisms (Supplementary Data 29 and Fig. 5). Since the MR analysis could not be applied to the larger, better-powered POAG cross-ancestry GWAS, as it requires a similar population background between the e/sQTL and GWAS studies (European in our case), our downstream analyses were applied to the more inclusive list of proposed causal genes based on the colocalization analysis (Supplementary Data 13).

### Enrichment of POAG and IOP colocalizing e/sGenes in biological processes

To gain biological insight into ways the implicated genes might contribute to glaucoma pathogenesis, we next tested whether the target genes of all the colocalizing e/sQTLs with POAG cross-ancestry, POAG EUR, or IOP GWAS loci were enriched in specific biological pathways, gene ontologies, or mouse phenotype ontologies, using *GeneEnrich* (Methods and Fig. 1c). Genes that colocalized with POAG cross-ancestry loci were significantly enriched in elastic fiber formation (Empirical P-value (P) < 1 × 10^{-5}, FDR < 0.001) and extracellular matrix organization (P = 3 × 10^{-5}, FDR = 0.012), and nominally enriched (P < 0.05) in the transforming growth factor beta (TGF) receptor signaling pathway (P = 3 × 10^{-4}) and abnormal eye morphology (P = 2.6 × 10^{-3}), amongst others (Supplementary Data 30 and Supplementary Fig. 14a,b). Genes that colocalized with POAG EUR loci were nominally enriched (P < 4 × 10^{-3}) in cellular senescence and cell cycle processes (e.g., Cyclin D-associated events in G1), lipid-related processes, such as apolipoprotein binding and decreased circulating high-density lipoprotein cholesterol level, and retina or neuronal related processes, including abnormal retina morphology, abnormal sensory neuron innervation pattern, and negative regulation of axon extension involved in axon guidance (Supplementary Data 31 and Supplementary Fig. 14c, d).

For the IOP genes, significant enrichment (P = 2 × 10^{-5}, FDR = 0.025) was found in transcriptional regulation by *VENTX*, a gene that encodes a homeodomain-containing transcription factor (Supplementary Data 32 and Supplementary fig. 14e). *VENTX* and its IOP-colocalizing target genes driving the gene set enrichment signal (*ANAPC1, ANAPC7, AGO4, MOV10, TCF7L2*) were most highly expressed in immune cell types, lymphocytes and macrophages, in the single cell anterior segment and optic nerve head described below (Supplementary Fig. 14f). The IOP genes were also significantly enriched (FDR < 0.15) in blood vessel morphogenesis and vasculature development, regulation of cytoskeletal organization, negative regulation of

cellular component organization, and adherens junction (Supplementary Data 32 and Supplementary Fig. 14e, g). Since colocalization with multiple e/sQTLs was found for two GWAS loci in the HLA region on chromosome 6 associated with POAG and IOP (29 and 35 e/sGenes, respectively), likely due to high LD in the HLA region, we removed this region from the gene set enrichment analysis above to avoid inflating the results due to a single locus. When kept in, the endosomal vacuolar pathway, interferon gamma signaling, antigen presentation folding assembly and peptide loading of class I MHC, negative regulation of natural killer cell-mediated immunity, and cell aging were significantly enriched (FDR < 0.1) for POAG genes (Supplementary Data 33), in addition to the gene sets above. The colocalizing POAG and IOP genes driving the gene set enrichment signals are listed in Supplementary Data 30–33.

### Identifying pathogenic cell types for POAG and related ocular traits

To further relate the implicated genes to pathogenic mechanisms and cell types, we next tested whether the expression of POAG or IOP colocalizing e/sGenes was enriched in specific cell types in key eye tissues implicated in the pathophysiology of POAG. We first applied ECLIPSER[38,39] (Methods and Fig. 1d) to 228, 118, and 279 e/sGenes that colocalized with POAG cross-ancestry, POAG EUR and IOP GWAS loci, respectively (Supplementary Data 13 and 34), and to cell type-specific expression from single nucleus (sn) RNA-seq of 13 tissues dissected from non-diseased human eyes: central cornea, corneoscleral wedge (CSW), trabecular meshwork (TM) including Schlemm's canal, iris, ciliary body (CB), lens[36] (all from anterior segment), peripheral and macular retina[33,36], the optic nerve head (ONH), optic nerve (ON), peripapillary sclera (PPS), peripheral sclera, and choroid[37] (all from posterior segment). The cell type enrichment results are summarized in Supplementary Data 35 and Table 2.

In the anterior segment, we found significant enrichment (tissue-wide FDR < 0.1) for POAG EUR loci in fibroblasts derived from the ciliary muscle (present in CB, CSW and TM[36]), annotated as ciliary fibroblasts in van Zyl et al.[36], followed by fibroblasts derived from the iris root (present within the iris[36]), annotated as iris fibroblasts (Supplementary Data 35 and Fig. 6a, b). The ciliary muscle and iris are key tissues involved in the unconventional outflow pathway. These fibroblasts were also detected histologically within the TM where all three tissues meet and interweave at the iridocorneal angle[36], implicating the conventional aqueous outflow pathway as well. Figure 6c shows the e/sGenes driving the POAG EUR enrichment signal in ciliary fibroblasts, also contained among the POAG cross-ancestry genes enriched in ciliary fibroblasts (Supplementary Fig. 15a, b). The POAG EUR genes were also modestly enriched in fibroblasts derived predominantly from the TM tissue (annotated as TM fibroblasts[36]) (P = 0.014, FDR = 0.18). For POAG cross-ancestry and IOP loci, we found supportive enrichment (P < 0.05, FDR < 0.23) in outflow pathway and cornea fibroblasts, vascular endothelium cells (cluster 2 derived from TM, CSW and CB tissues[36]), and lens epithelium, as detailed in Figs. 6a, d and 7a and Supplementary Data 35. Genes that colocalized with IOP loci were also significantly enriched (FDR < 0.1) in pericytes (cluster 2 that localizes to the CSW[36]), and nominally enriched in lymphatic endothelium and Schlemm's canal, whose

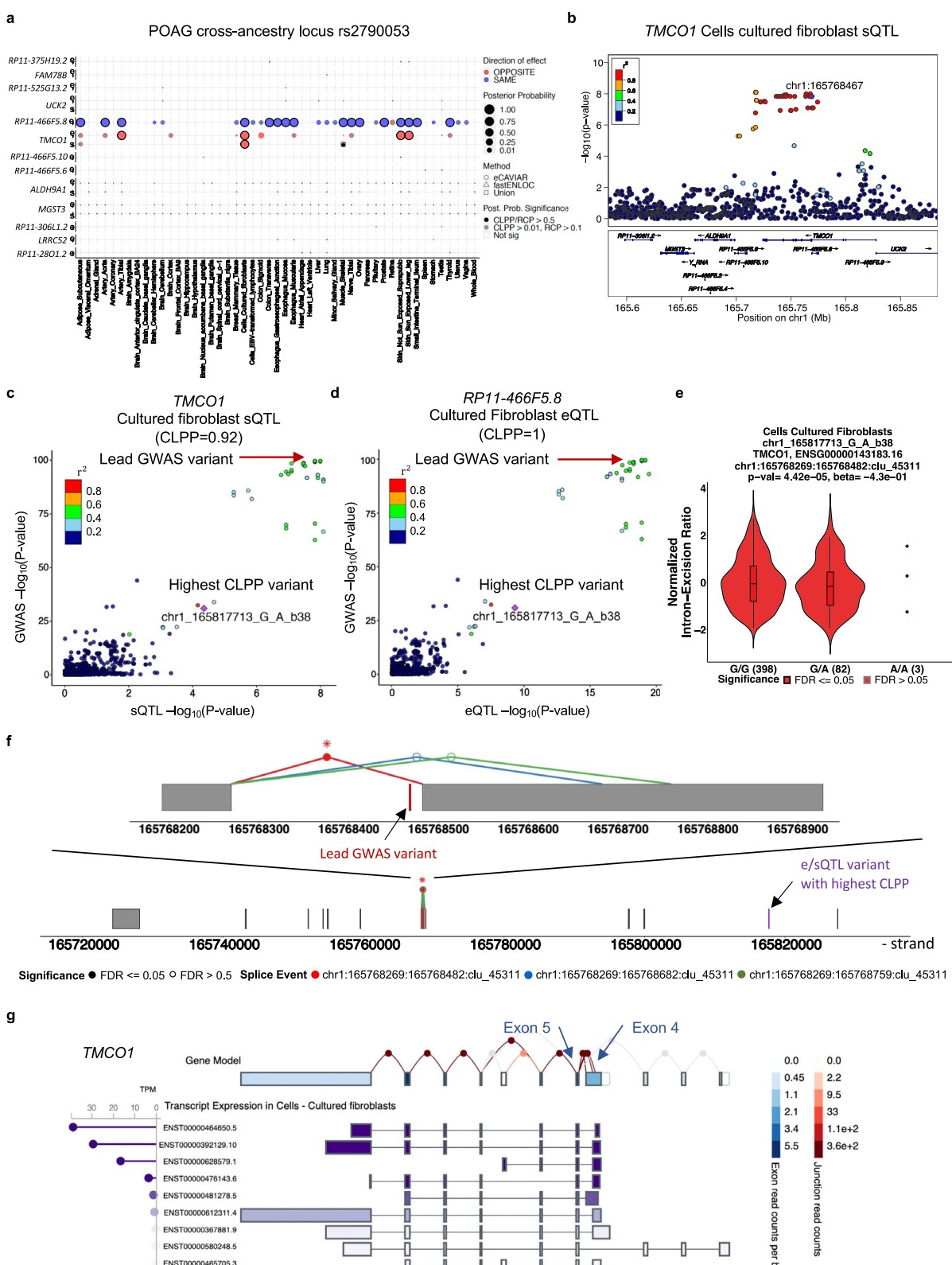

dysfunction can lead to elevated IOP[49] (Fig. 7a and Supplementary Data 35).

Clustering of the top-ranked anterior segment cell types (P < 0.05) for POAG and IOP, separately, based on the overlap of genes driving the cell type enrichment, suggests three cell classes affecting POAG - fibroblasts, vascular endothelium, and lens epithelium. and three cell classes for IOP - fibroblasts, pericytes, and lymphatic endothelial cells

(Figs. 6e and 7b). Between 45-78% of the genes driving the POAG enrichment signals in the outflow fibroblast cell types are common between the different fibroblasts (Fig. 6e), suggesting both shared and distinct genes acting in the conventional and unconventional outflow pathways. The IOP genes driving the enrichment signal in pericytes (Fig. 7c, d) were largely distinct from those enriched in vascular and fibroblast cell types (overlap 7-33%. Fig. 7b), and were enriched in

**Fig. 4 | Example of colocalizing e/sQTLs with a top POAG and IOP GWAS locus.**
**a** Colocalization results for all eQTLs (e) and sQTLs (s) across 49 GTEx tissues and retina overlapping the POAG cross-ancestry rs2790053 LD interval, using eCA-VIAR and *enloc*. Genes are ordered by chromosome position. Point size is proportional to maximum colocalization posterior probability (CLPP or RCP) of all e/sVariants tested per gene-QTL-tissue combination. Points are color-coded by direction of effect (blue if increased expression or splicing increases POAG risk or vice versa; red if increased expression or splicing decreases POAG risk or vice versa). Circle: eCAVIAR, triangle: *enloc*, and square: tested in both methods; results shown for method with maximum posterior probability. Grey or black border denote variant-gene-tissue-QTL combination that passed colocalization posterior probability above 0.01/0.1 (CLPP/RCP) or 0.5, respectively, and QC filtering (Methods). White or black asterisk in the square indicates whether the second method passed CLPP > 0.01/RCP > 0.1 or CLPP/RCP > 0.5, respectively.
**b** LocusZoom[96] plot for *TMCO1* sQTL -log$_{10}$(*P*-value) in GTEx Cells-Cultured fibroblasts in POAG cross-ancestry GWAS rs2790053 (chr1_165768467_C_G_b38) LD interval. Points color-coded by LD (r$^2$) relative to lead GWAS variant,

chr1_165768467_C_G_b38. LocusCompare plots of POAG **c**ross-ancestry GWAS meta-analysis -log$_{10}$(*P*-value) versus -log$_{10}$(*P*-value) of *TMCO1* sQTL (**c**) or *RP11-466F5.8* eQTL (**d**) in Cells-Cultured fibroblasts. Points color-coded based on LD (r$^2$) relative to e/sVariant with highest CLPP. **e** Violin plot of normalized intron-excision ratio at chr1:165768269-165768482 (Leafcutter[95]) for *TMCO1* in fibroblasts versus genotype of sVariant chr1:165817713_G_A_b38 with highest CLPP for POAG cross-ancestry locus rs2790053. Boxplots represent median and inter-quartile range (IQR); whiskers mark 1.5x IQR; violin plot edges represent minima and maxima. **f** *TMCO1* gene model (grey boxes: exons) in GTEx fibroblasts showing all intron excision splicing events detected with Leafcutter; zoom-in: alternative splice donor site events on exon 4 whose sQTL colocalized with POAG (red versus blue or green). **g** Exon boxes in *TMCO1* gene model color-coded by exon read counts per base (blue) in GTEx Cells-Cultured fibroblasts; lines connecting exons for all splicing events are color-coded by exon-exon junction read counts (red). Transcript expression shown in Transcripts per Million (TPM) from RSEM[112]. Plots are taken from https://gtexportal.org/.

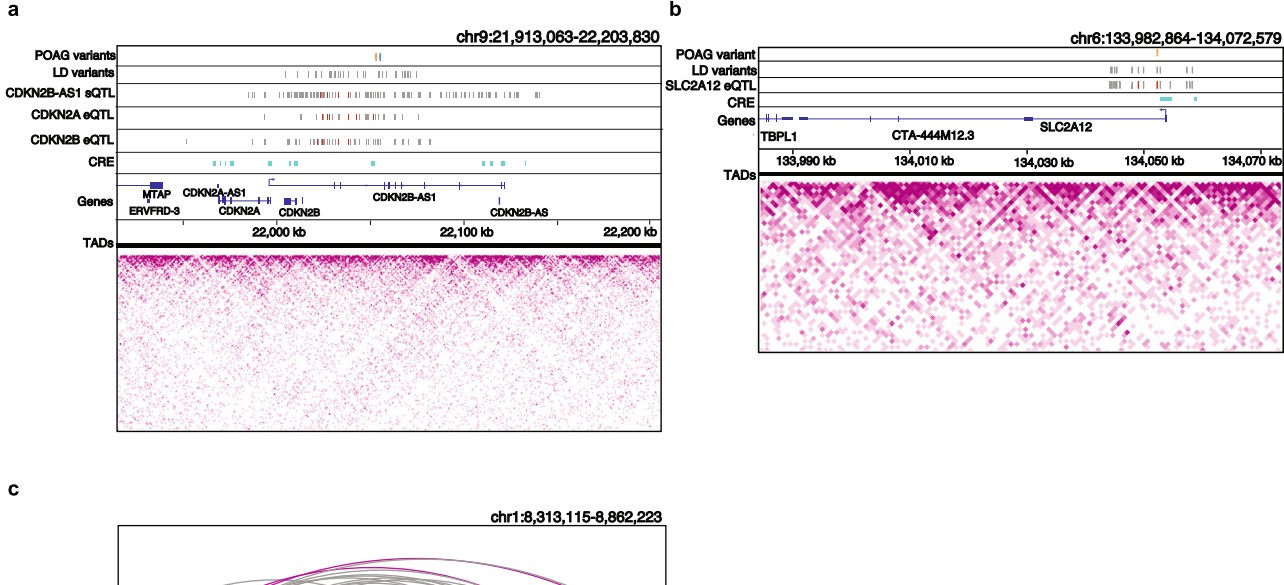

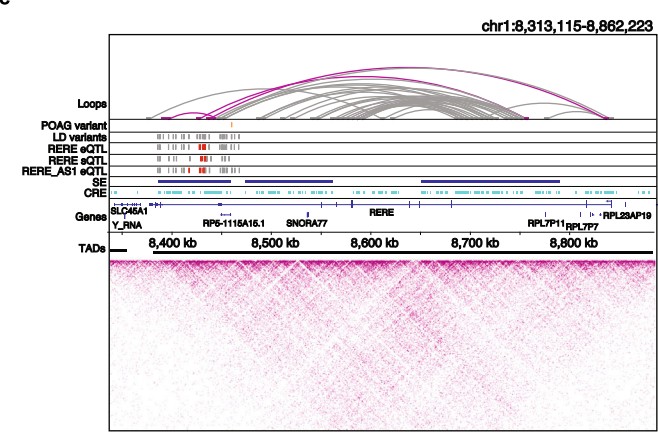

**Fig. 5 | Chromatin loops and regulatory elements in human retina support effect of colocalizing e/sQTLs on POAG risk. a** Retina CREs (cyan) derived from epigenetic data overlapping e/sVariants that colocalized with POAG associations in the *CDKN2A/B* locus. The lead POAG variants from the cross-ancestry (blue line) and European subset (orange line) GWAS are shown in the top track, followed by their linkage disequilibrium (LD) proxy variants (r$^2$ > 0.8) in the track below. The significantly colocalizing *CDKN2B-AS1* sVariants in Pituitary, *CDKN2A* eVariants in Brain Cortex, and *CDKN2B* eVariants in Skeletal Muscle are represented by red lines, and the grey lines represent LD proxy variants to the colocalizing e/sVariants, which are also significant e/sQTLs (FDR < 0.05) for the corresponding gene and tissue.
**b** Retina CREs (cyan) overlapping retina *SLC2A12* eVariants that colocalized with the POAG cross-ancestry association. Tracks display the lead POAG cross-ancestry GWAS variant rs2811688 (orange) and its LD proxy variants (grey), followed by the significantly colocalizing *SLC2A12* retina eVariants (red) with their LD proxy

eVariants, which are also significant eQTLs at FDR < 0.05 (grey). The CRE overlaps the promoter of *SLC2A12*. **c** Retina chromatin loops from Hi-C (3D chromosome conformation capture) data, SEs (blue), and CREs (cyan) shown for the *RERE* POAG locus. Tracks display the lead POAG cross-ancestry GWAS variant rs172531 (orange) and its LD proxy variants (grey), followed by significantly colocalizing *RERE* eVariants in Nerve Tibial, *RERE* sVariants in Cell-Cultured Fibroblast cells, and *RERE-AS1* eVariants in Adipose Subcutaneous (red), and their LD proxy variants that are also significant e/sQTLs (FDR < 0.05) for the corresponding gene and tissue (grey). Magenta loops have one foot that overlaps or is in LD with the POAG variant and colocalizing e/sQTLs. In all panels, LD proxy variants were computed at r$^2$ > 0.8, TADs are represented as solid black lines, and the magenta heatmaps represent Hi-C physical contact maps. CRE *Cis*-regulatory element, SE Super-enhancer, TAD Topologically associating domain.

**Table 2 | Summary of ocular cell types enriched for e/sQTL-mapped genes in POAG and/or IOP GWAS loci**

| Tissue | POAG-specific | IOP-specific | POAG and IOP |
|---|---|---|---|
| Anterior segment | - | Pericytes<br>Schlemm's Canal endothelium* | Ciliary muscle fibroblasts<br>Trabecular meshwork fibroblasts<br>Iris fibroblasts<br>Lens epithelium<br>Vascular endothelium<br>Lymphatic endothelium |
| Retina | Astrocytes<br>Müller Glia | – | – |
| Optic nerve head & surrounding posterior tissues | Oligodendrocytes (ON, ONH)<br>Oligodendrocyte precursor cells (ON, ONH)<br>Astrocytes (ONH, ON, PPS) | Vascular endothelium (choroid, ONH, PPS, sclera)<br>Vascular smooth muscle cells (PPS, sclera)<br>Pericytes (ONH, choroid, PPS) | Fibroblasts (choroid, ONH, PPS, sclera)<br>Nonmyelinating Schwann cells (choroid, PPS) |

This table lists ocular cell types significantly enriched for cell type-specific expression of genes mapped to POAG and/or IOP GWAS loci based on e/sQTLs, according to ECLIPSER analysis of all or unique POAG and IOP loci at tissue-wide Benjamini-Hochberg FDR < 0.1 (Supplementary Data 35 and 38). *Nominal enrichment ($P < 0.05$). POAG results are summarized from the POAG cross-ancestry and European GWAS meta-analyses. The posterior tissues in which the enriched cell types are most abundant are listed in parentheses. *ONH* optic nerve head, *ON* optic nerve, *PPS* peripapillary sclera.

vasculature development ($P = 3 \times 10^{-5}$, FDR = 0.075; Supplementary Data 37). On the other hand, the IOP genes driving the enrichment in TM fibroblasts (Supplementary Fig. 15c,d) were highly shared with the IOP genes enriched in ciliary and iris fibroblasts (overlap 64–88%. Fig. 7b). Notably, the enrichment of IOP genes in pericytes was specific to IOP (asterisks in Fig. 7c). When ECLIPSER was applied to genes that colocalized with IOP loci not associated with POAG, only the enrichment in pericytes remained ($P = 0.007$) (Supplementary Data 38 and Supplementary Fig. 15e). Genes mapped to shared IOP and POAG loci were significantly enriched in ciliary and TM fibroblasts (FDR < 0.01) and lymphatic or vascular endothelial cells (FDR = 0.026). No enrichment was found for POAG-only loci in the anterior segment cell types, supporting IOP-dependent mechanisms in the anterior segment for POAG risk, as expected (Supplementary Data 38).

We next tested for enrichment of POAG and IOP colocalizing e/sGenes in retina snRNA-seq data (Methods). We found significant enrichment of POAG cross-ancestry genes in astrocytes and Müller glia cells (FDR < 0.04; Supplementary Data 35, Fig. 8a and Supplementary Fig. 16a), which replicated (FDR < 0.06) in a separate snRNA-seq study of the macula (Methods; Supplementary Data 35 and Supplementary Fig. 16g). Consistent results were found for POAG EUR genes (Supplementary Fig. 16b). A quarter (*YAP1, LPP, TRIB2)* of the 12 POAG cross-ancestry genes driving the astrocyte enrichment were common with Müller glia cells (Supplementary Fig. 16d, e), suggesting both shared and distinct processes between the two cell types. IOP genes were only nominally enriched in astrocytes ($P = 0.032$; Supplementary Fig. 16c). By testing POAG- or IOP-only loci and shared loci, the POAG enrichment in retinal astrocytes and Müller glia cells appears to be independent of IOP (Supplementary Data 38 and Supplementary Fig. 17; more details in Supplementary Note 6). The POAG cross-ancestry genes were also enriched in RPE cells and S-cones in the macula (FDR = 0.06). Of note, no significant enrichment was observed in RGCs (Fig. 8a), though some POAG colocalizing e/sGenes are expressed in RGCs (Supplementary Fig. 20).

Finally, we tested for cell type-specific enrichment in the optic nerve head (ONH), optic nerve (ON) and adjacent posterior tissues (Methods). The strongest enrichment (FDR < 0.01) of POAG cross-ancestry genes was found in fibroblasts primarily in the peripapillary sclera (PPS) that encompasses the ONH, followed by fibroblasts most abundant in the choroid, astrocytes that reside in the ONH and ON, Schwann cells in the choroid and PPS, oligodendrocyte precursor cells (OPCs) and oligodendrocytes in the ON and ONH (Supplementary Fig. 18f, g), and vascular endothelium cells primarily in the choroid (FDR < 0.09; Fig. 8b, c, Supplementary Data 35 and Supplementary Fig. 18a,d). POAG EUR genes showed similar enrichment patterns (Fig. 8b, Supplementary Fig. 18b and Supplementary Data 35). About

half the genes driving the enrichment in astrocytes in ONH (Fig. 8d, e) and retina samples from separate donors were common (e.g., *DGKG, PLCE1, LPP, GAS7, YAP1*, and *COL11A1*; Supplementary Data 35). *DGKG*, diacylglycerol kinase gamma, whose retina-specific eQTL colocalized (CLPP = 0.96) with POAG cross-ancestry association (Fig. 8f) displayed the strongest astrocyte-specificity in ONH (Fig. 8d) and retina (Supplementary Fig. 16d), compared to all other astrocyte-specific POAG-colocalizing e/sGenes.

IOP genes were most significantly enriched in vascular endothelial cells and fibroblasts primarily residing in the choroid, but also in the ONH and PPS (FDR < 0.014), followed by Schwann cells in the choroid and PPS, vascular smooth muscle cells (19-ACTA2; Supplementary Fig. 18h,I, and 34-ACTA2) in the PPS and sclera, pericytes (26-ACTA2) in the ONH, PPS and choroid, and OPCs in the ON (FDR < 0.06; Fig. 8b, Supplementary Data 35, and Supplementary Fig. 18a, c, e). The IOP genes driving enrichment in vascular endothelial cells in the ONH, choroid and posterior tissues were enriched in vasculature development and anchoring junction gene ontologies (FDR < 0.06), and IOP genes enriched in pericytes in the ONH, choroid and PPS were enriched in TIE2 signaling (FDR = 0.02), response to carbohydrate adhesion (FDR = 0.13), and negative regulation of cell adhesion (FDR = 0.13) (Supplementary Data 37). Notably, the enrichment in oligodendrocytes, OPCs and astrocytes was specific to POAG-only loci, and enrichment in vascular endothelium and mural cells was specific to IOP-only loci (Supplementary Data 38 and Supplementary Fig. 19).

The cell type expression profiles of all POAG cross-ancestry, POAG EUR and IOP colocalizing e/sGenes is shown in Supplementary Fig. 20, and a summary of the cell types and pathways in which each of the POAG and IOP colocalizing e/sGenes are enriched is presented in Supplementary Data 39–41. Applying ECLIPSER to various negative control traits suggests that the cell type enrichment results are specific to glaucoma and not due to unaccounted confounding factors (Supplementary Data 42, 43, Fig. 6a, Supplementary Fig. 21, and Supplementary Note 7). Furthermore, the ECLIPSER cell type enrichment significance did not correlate with cell count per cell type in the single-nucleus datasets (Pearson's $R^2 < 0.2$, $P > 0.12$; Supplementary Data 44).

To increase confidence in the POAG and IOP cell type enrichment results, we applied two additional methods that identify cell types associated with complex traits, through regression analysis of genome-wide associations beyond known GWAS loci: stratified LD score regression (S-LDSC) and MAGMA (Fig. 1e; Methods). The primary enriched cell types for POAG and IOP found with ECLIPSER, including ciliary and TM fibroblasts, ONH fibroblasts, and retinal macroglial cells, were significant with S-LDSC (Supplementary Data 45 and Supplementary Fig. 22) and more restrictively with MAGMA (Supplementary Data 46 and Supplementary Fig. 22); additional enrichment was found

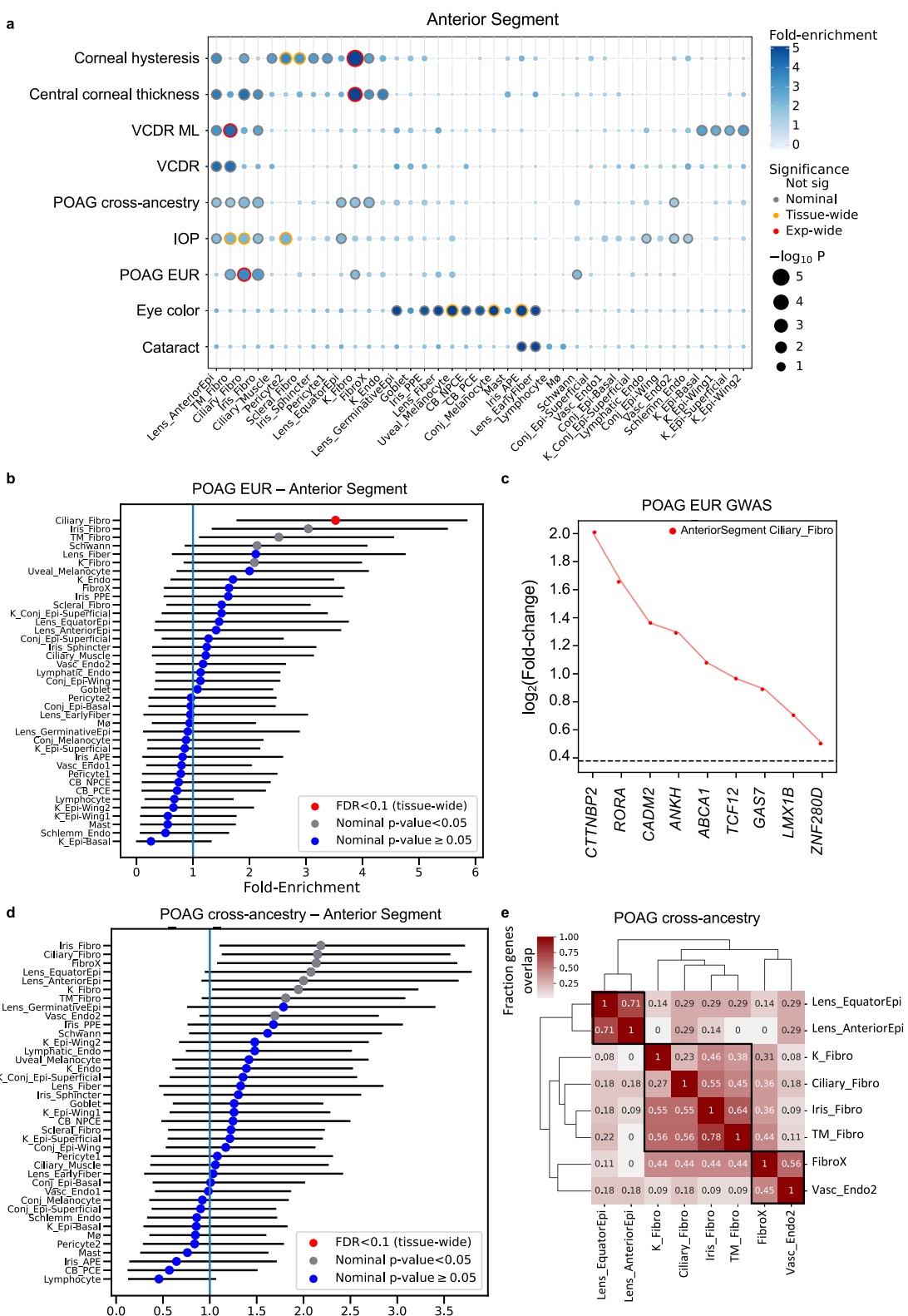

in vascular types. Enrichment of POAG loci in ONH and ON oligodendrocytes was only found with ECLIPSER suggesting that the enrichment is primarily driven by genes with strong genetic effects. The cell type enrichment significance of ECLIPSER was reasonably correlated with that of S-LDSC and MAGMA (Average Pearson's $r = 0.53$, range: 0.18-0.86; Supplementary Data 47). We further used conditional analysis implemented in MAGMA to test whether the different cell type enrichment signals for POAG or IOP were independent of each other in

each tissue (Supplementary Data 48). In the anterior segment, the enrichment of POAG associations in ciliary fibroblasts was independent of TM fibroblasts, but not vice versa (Conditional $P = 0.04$). In retina, POAG associations were significantly enriched in astrocytes ($P < 6E-8$) and Müller glia cells ($P < 0.002$), but only astrocytes remained significant after conditional analysis (Conditional $P < 4E-6$), suggesting that astrocytes may play a more important role in glaucoma pathogenicity than Müller Glia cells. In the ONH, the POAG and IOP

**Fig. 6 | Cell type enrichment of e/sQTL-mapped genes to POAG, IOP and related trait GWAS loci in the ocular anterior segment. a** Significance (circle size, -log10 (*P*-value)) and fold-enrichment (circle color) of the cell type specificity of e/sGene mapped-GWAS locus sets for POAG cross-ancestry, POAG European (EUR) subset, intraocular pressure (IOP), central cornea thickness, corneal hysteresis, physician-defined vertical-cup-to-disc ratio (VCDR), and machine learning-defined VCDR (VCDR ML), and as positive controls, eye color and cataract, using ECLIPSER (one-sided test), shown for 39 cell types in six ocular anterior segment tissues. Traits (rows) and cell types (columns) were clustered based on hierarchical clustering of Euclidean distance between the cell type-specificity enrichment scores of GWAS locus sets. Red rings: experiment-wide significant (Benjamini-Hochberg (BH) FDR < 0.1). Yellow rings: tissue-wide significant (BH FDR < 0.1). Grey rings: nominal significant (*P* < 0.05). **b, d** Cell type specificity fold-enrichment (x-axis) in the anterior segment cell types ranked in descending order for e/sQTL-mapped genes to GWAS loci of the POAG European subset (No. loci (*N*) = 37) (**b**) and POAG cross-ancestry (*N* = 79) (**d**). Points: fold-enrichment estimates from ECLIPSER. Error bars:

95% confidence intervals. Red: tissue-wide significant (BH FDR < 0.1). Grey: nominal significant (*P* < 0.05). Blue: non-significant (*P* ≥ 0.05). The exact *p*-values (one-sided) for the fold-enrichment of all cell types and GWAS are provided in Supplementary Data 35. **c** Differential gene expression (log₂(Fold-change), y axis) in the most strongly enriched cell type compared to all other cell types is shown for the set of genes (x axis) driving the enrichment signal of the POAG European GWAS loci in ciliary fibroblasts. The horizontal dashed line represents $\log_2$(Fold-change) of 0.375 (FC = 1.3) and FDR < 0.1 used as the cell type-specificity enrichment cutoff. **e** Heatmap of fraction of genes that overlap between the e/sGenes driving the enrichment signal for top ranked cell types (*P* < 0.05) in the anterior segment for POAG cross-ancestry GWAS loci. Numbers refer to fraction of e/sGenes driving the cell type enrichment on each row that overlaps with the genes driving the cell type enrichment on the corresponding column. Hierarchical clustering was performed on both rows and columns using Euclidean distance between fractions. Cell type abbreviations are described in Supplementary Data 35.

enrichment in fibroblasts, astrocytes, vascular endothelium, and mural cells were all independent of each other (Supplementary Data 48).

Finally, to augment the POAG and IOP enrichment analysis, we tested for cell type enrichment of genes mapped to GWAS loci of additional glaucoma associated traits (listed in Supplementary Data 34), including vertical-cup-to-disc ratio (VCDR), cornea hysteresis, and central cornea thickness (Methods) in all ocular tissue regions (Supplementary Data 35, Fig. 6a and Fig. 8a, b). In the anterior segment, genes mapped to central corneal thickness and corneal hysteresis were most significantly enriched in corneal fibroblasts (FDR < 0.007; Fig. 6a and Supplementary Fig. 23a, b), highlighting the specificity of the POAG and IOP gene enrichment in the outflow pathway fibroblasts. The VCDR GWAS loci from a well-powered GWAS that used deep learning (ML) to score the characteristics of fundus images from 65,680 European individuals[12] showed significant enrichment in the TM fibroblasts from the conventional outflow pathway (FDR = 0.02; Fig. 6a and Supplementary Fig. 23c). TM fibroblasts were also the top nominally enriched cell type for a smaller VCDR GWAS, where 23,899 fundus images were manually scored by ophthalmologists[13] (P = 0.0076, FDR = 0.3; Fig. 6a and Supplementary Fig. 23d). In the retina, the ML-based VCDR loci displayed significant enrichment in GABAergic amacrine cells, cone photoreceptors and Müller glia cells (FDR < 0.085; Fig. 8a and Supplementary Fig. 23e) and nominal enrichment in astrocytes. In the ONH, the ML-based VCDR genes were nominally enriched in fibroblasts in the PPS, vascular endothelium primarily in the choroid, and astrocytes in the ONH and ON, similarly to POAG loci (Fig. 8b, Supplementary Fig. 23g and Supplementary Data 35).

In summary, our cell type enrichment analysis has revealed roles for both known and less well-studied cell types in POAG pathogenicity, such as fibroblasts in the unconventional and conventional outflow pathways, astrocytes in retina and ONH, OPCs in the ON and ONH, and Schwann cells and fibroblasts in the PPS and choroid (Table 2). It also suggests known and new causal genes for POAG and related eye traits that may be affecting glaucoma susceptibility through specific cell types in the anterior and posterior parts of the eye in IOP-dependent and independent manners (Supplementary Data 35 and 38).

## Discussion

We report results of a systematic investigation of the underlying causal mechanisms, genes and cell types of over 130 cross-ancestry or European loci associated with POAG[8,9] and over 110 loci associated with its major risk factor, elevated IOP[9]. Our analysis integrated a variety of datasets, including expression and splicing QTLs from 49 GTEx tissues[26] and from retina[27], genome topology data from retina[40], single-nucleus expression data from a whole-eye cell atlas that includes key structures of both the anterior[36] and posterior segments[33,37], and the largest to date GWAS meta-analyses for these traits[8,9]. Our finding that

eQTLs and sQTLs in GTEx tissues and retina are enriched for hundreds of known and more modest POAG and IOP associations, suggests a primary role for transcriptional regulation in POAG susceptibility, as observed for other diseases[24–27], and implies that GTEx tissues can be used to uncover causal mechanisms for glaucoma. The GTEx e/sQTLs likely capture shared genetic regulation with the actual pathogenic tissues and cell types for glaucoma, such as fibroblasts and vascular endothelial cells in the outflow pathways and ONH region, due to shared regulation across cell types and tissues[26,38].

Using two QTL/GWAS colocalization methods[41,42], we prioritized putative causal genes for ~60% of the POAG and IOP GWAS loci. A similar fraction of GWAS loci with significant colocalization results has been found for other complex diseases and traits[26,50]. For a quarter (80) of the POAG and IOP loci, a single gene was proposed, ten of which are noncoding genes (lincRNA and antisense), suggesting that transcriptional and post-transcriptional gene regulation contribute to glaucoma susceptibility. We provided additional support for three quarters of the colocalizing e/sGenes using Mendelian randomization[43,51,52], which tests for horizontal pleiotropy, not accounted for by Bayesian colocalization analysis[53,54]. For about one third of the GWAS loci, none of the proposed causal gene/s were the nearest gene to the lead GWAS variant, similar to that observed for other complex traits[24,50,55]. These results emphasize the value of using e/sQTLs or other functional assays that link regulatory regions to distal target genes[24,56–58] to prioritize causal genes underlying common variant associations.

Integrating e/sQTLs with POAG and IOP GWAS loci proposed both previously suggested[8] and new biological processes for these traits. The POAG colocalizing genes, which included several known mendelian, early-onset glaucoma genes (*EFEMP1*[47,59] and *LTBP2*[60]), were most strongly enriched in extracellular matrix organization and elastic fiber formation, as previously reported[8], followed by TGF receptor signaling pathway. Structural changes of the extracellular matrix induced by TGF-beta2 in both the trabecular meshwork in the outflow pathway and the optic nerve head have been associated with POAG[61], and have been suggested to cause impairment of optic nerve axonal transport and neurotrophic supply that could influence RGC degeneration[61]. Regulation by the homeodomain transcription factor, VENTX, was the most significantly enriched gene set for IOP genes, which has not yet been associated with glaucoma. VENTX is proposed to play important roles during embryonic patterning (by homology), including in neural crest development[62], as well as hematopoiesis, leukemogenesis, cellular senescence and macrophage differentiation[63]. Its strongest expression in our single cell data was in lymphocytes in the anterior segment, and macrophages in the optic nerve head, proposing a novel link between immune-related processes and IOP levels. Reduced circulating endothelial progenitor cells has been reported in POAG patients[64], which could

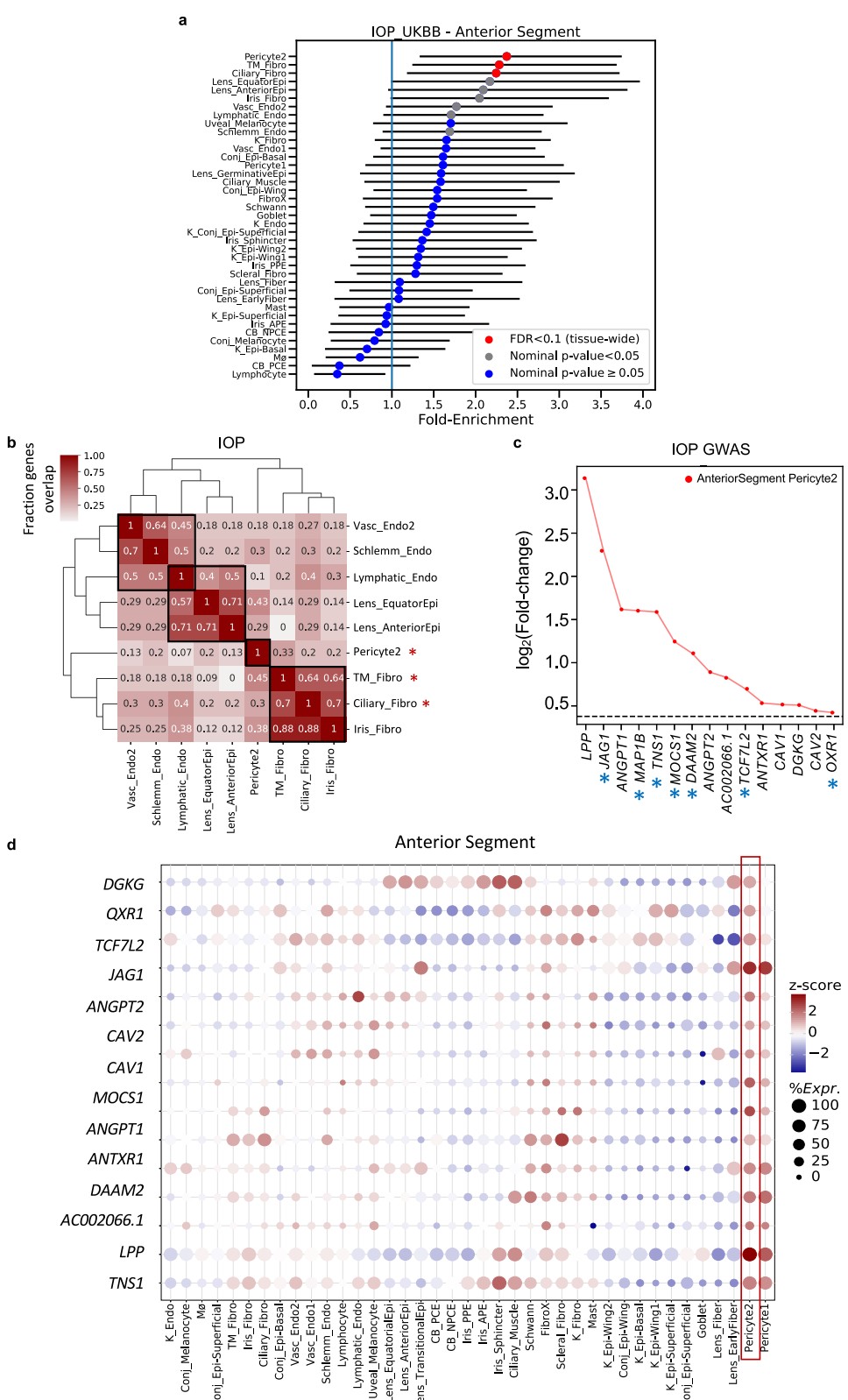

explain impaired flow-mediated vasodilation in POAG[65]. Additional processes suggested to affect IOP regulation, aside from previously suggested[9] vascular development, are regulation of cytoskeleton organization, and adherens junction, a cell-cell junction whose cytoplasmic face is linked to the actin cytoskeleton, and that allows cells to respond to biomechanical forces and structural changes in the tissue microenvironment[66]. Inhibition of adherens junction

regulation in trabecular meshwork has been shown to modestly influence IOP levels in rabbits[67]. We also found modest enrichment of POAG genes in neuronal-related processes, including genes affecting retinal morphology, sensory neuron innervation pattern, and regulation of axon guidance. These genes may represent IOP-independent mechanisms, which will need to be corroborated in future GWAS with larger numbers of normal tension glaucoma cases.

**Fig. 7 | Cell type enrichment of e/sQTL-mapped genes to IOP GWAS loci in the anterior segment of the eye. a** Cell type specificity fold-enrichment (x-axis) in the anterior segment cell types ranked in descending order for e/sQTL-mapped genes to IOP GWAS loci (*N* = 37). Points: fold-enrichment estimates from ECLIPSER. Error bars: 95% confidence intervals. Red: tissue-wide significant (Benjamini-Hochberg (BH) FDR < 0.1). Grey: nominal significant (*P* < 0.05). Blue: non-significant (*P* ≥ 0.05). The exact *p*-values (one-sided) for the fold-enrichment of all cell types and GWAS are provided in Supplementary Data 35. **b** Heatmap of fraction of genes that overlap between the e/sGenes driving the enrichment signal for top ranked cell types (*P* < 0.05) in the anterior segment for IOP GWAS loci. Numbers refer to fraction of e/sGenes driving the cell type enrichment on each row that overlaps with the genes driving the cell type enrichment on the corresponding column. Hierarchical clustering was performed on both rows and columns using the Euclidean distance between fractions. Red asterisks: tissue-wide cell type enrichment BH FDR < 0.1

from ECLIPSER. **c** Differential gene expression (log$_2$(Fold-change), *y* axis) in the most strongly enriched cell type in the anterior segment compared to all other cell types is shown for the set of genes (x axis) driving the enrichment signal of IOP GWAS loci in pericytes (cluster 2). The horizontal dashed line represents log$_2$(Fold-change) of 0.375 (FC = 1.3) and FDR < 0.1 used as the cell type-specificity enrichment cutoff. Blue asterisks denote genes in IOP loci not associated with POAG risk. **d** Bubble map displaying the expression of e/sGenes driving the IOP enrichment in pericytes (red box) across all cell types in the anterior segment. The colorbar represents gene expression z-scores computed by comparing each gene's average expression in a given cell type to its average expression across all cell types divided by the standard deviation of all cell type expression averages. Bubble size is proportional to the percentage of cells expressing the given gene (log(TPK + 1) > 1). Cell type abbreviations are described in Supplementary Data 35.

In addition to prioritizing causal genes for POAG and IOP, e/sQTLs suggest the direction of effect of gene expression changes or alternative splicing on disease risk that could inform drug design. In this study, we provide various hypotheses of putative causal genes and regulatory mechanisms that may affect POAG susceptibility in an IOP-dependent or independent manner, for experimental follow up. For example, an increase in expression or alternative splicing of *TMCO1*, a gene that regulates the balance of calcium ions inside the endoplasmic reticulum, or a decrease in expression of *LMX1B*, LIM homeobox transcription factor 1 beta, that is essential for several developmental processes including the anterior segment of the eye[68], were proposed to reduce POAG risk and IOP levels. The lead IOP GWAS variant (rs116089225, a low frequency allele) in *TMCO1* that colocalized with *TMCO1* e/sQTLs has been recently associated with variable number tandem repeat (VNTR) length in the UK biobank study[69]. However, the VNTR did not display allelic series association with *TMCO1* expression levels in GTEx[69]. This and our finding that *TMCO1* e/sQTLs also colocalize with additional common, potentially less severe POAG and IOP risk variants in the locus suggest that the LD-independent GWAS variants in the locus might be tagging more than one causal mechanism. As for IOP-independent mechanisms, an sQTL acting on *CDKN2B-AS1*, which leads to skipping of exons 2 and 3 that overlap the *CDKN2B* gene on the opposite strand, is proposed as a potential mechanism of action for the protective signal found in this gene[70]. Skipping of these exons might render the *CDKN2B* antisense less efficient in forming a complex with the *CDKN2B* RNA. Retinal Hi-C and epigenetic data further support potential roles for e/sQTL effects on POAG in the retina, such as increased expression of *RERE* (arginine-glutamic acid dipeptide repeats) proposed to increase POAG risk. Of note, *RERE* has also been associated with VCDR[13]. Overexpression of the RERE protein that co-localizes with a nuclear transcription factor triggers apoptosis, and its deficiency in mice causes retinal and optic nerve atrophy[71]. This study also suggested a potential secondary causal gene, *PIGC* for the strongest POAG association, a nonsense mutation in *MYOC*, which will need to be replicated in a larger independent POAG GWAS. The *MYOC* mutation causes aggregation of misfolded myocilin proteins in the trabecular meshwork, which may lead to elevated IOP levels[72]. Conversely, the enrichment of *PIGC* (that encodes an endoplasmic reticulum-associated protein) expression, along with other POAG-colocalizing genes, in oligodendrocytes in the optic nerve head (ONH), suggests a secondary causal role for this locus in RGC support, in the posterior part of the eye.

There are several reasons why we may not have found colocalizing e/sQTLs for 40% of the loci. First, some of the causal genes or regulatory effects may be specific to regions in the eye or rare cell types for which we do not yet have representative e/sQTLs. Second, some genes may affect POAG or IOP by perturbing processes only active during development or under specific conditions or stimuli, not captured in adult tissues. Third, the causal variant may be another type of molecular QTL not tested in this study, such as e/sQTL acting in *trans*

or protein QTLs. Fourth, some of the genetic associations may be tagging deleterious protein-coding variants[8]. Finally, there are several limitations to Bayesian colocalization methodologies[53,54], as described in the Supplementary Discussion.

By applying a recently developed method (ECLIPSER[38,39]) to the colocalizing e/sGenes and single nucleus expression data from glaucoma-relevant eye tissues, we provided support for previously implicated cell types affecting POAG development, and shed light on less well-established or novel pathogenic cell types for POAG and IOP. One of the unique features of ECLIPSER, compared to other single cell enrichment methods[73–77] is that it identifies cell types that are specific to a given disease or trait, compared to a range of unrelated complex diseases and traits. Our results found that gene expression variation in the ciliary and iris fibroblasts in the unconventional outflow pathway in the anterior segment, in addition to the TM cells and Schlemm's canal cells in the conventional outflow pathway, may both be key contributors to local IOP regulation and POAG risk. The expression profile of the ciliary fibroblasts that was most strongly enriched for POAG genes, is most similar to the 'beam cell A' defined in the single cell RNA-seq atlas of the outflow pathway in van Zyl et al.[34] (Fig. S6B in[34]), which populates the ciliary muscle and uveal base of the TM. The iris fibroblasts are most similar to 'beam cell B', and the TM fibroblasts to the JCT (juxtacanalicular trabecular meshwork) cell type that resides adjacent to the Schlemm's canal. Since the role of the unconventional outflow pathway in IOP homeostasis remains relatively understudied compared to that of the conventional pathway[78], these findings may encourage further avenues of investigation. Furthermore, the enrichment of IOP genes in pericytes, many of which are not currently associated with POAG, extends our understanding of how genetic variation may be affecting IOP in the anterior segment. Pericytes are mural cells that wrap around the endothelial cells that line capillary blood vessels. A recent study has found reduced capillary diameter and impaired blood flow at pericyte locations in mouse eyes with high IOP[79].

In the peripheral and macular retina, we found significant enrichment of POAG colocalizing genes in astrocyte and Müller glia cells. Astrocyte and Müller glia are two types of macroglia cells that interact with RGCs and blood vessels, and play an important role in retinal homeostasis, including metabolic supply and structural support, maintaining the extracellular environment of the neurons, and neurotransmitter transmission. Müller glia, the most common glial cell in the retina, span the entire retinal layer, while astrocytes are present only in the innermost layer of the retina. Several studies in animal models and patients with glaucoma[80–82] have found that astrocytes and Müller glia cells become reactive at early stages of glaucomatous conditions when RGCs are intact, suggesting a role for macroglia in the initiation and progression of glaucoma. POAG genes, but not IOP genes, were also enriched in astrocyte types residing primarily in the ONH that contains the lamina cribrosa (LC)[83], a mesh-like structure where unmyelinated RGCs pass through the sclera to exit the eye, and

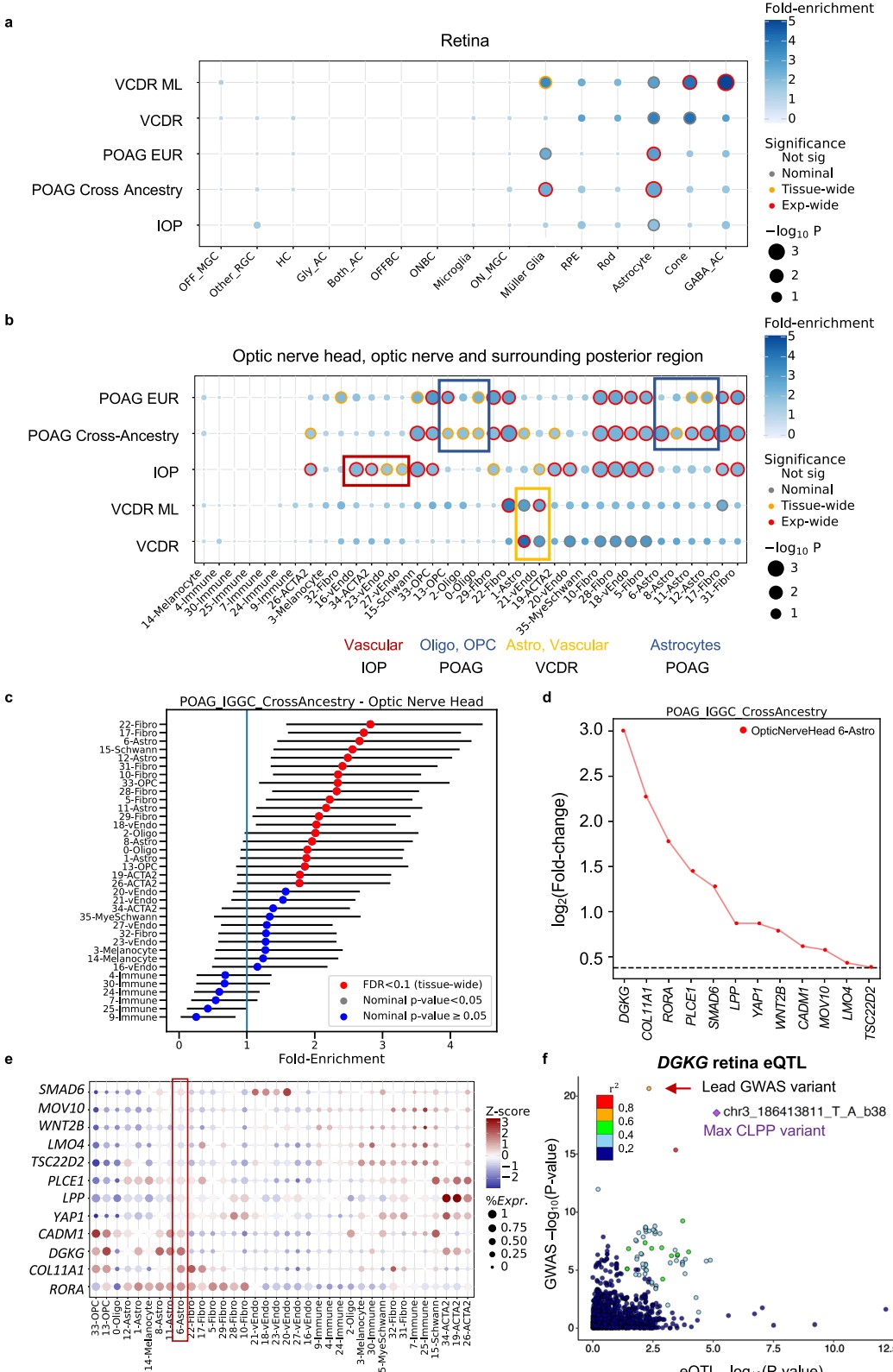

another in the ONH and optic nerve (ON). Notably, astrocytes have been found to be one of the major cell types isolated from human ONH[83] and in LC dissected from human ONH[84], and make up ~20% of the cells in our ONH snRNA-seq dataset.

In the ONH and surrounding posterior tissues, POAG genes were most strongly enriched in fibroblasts abundant in the peripapillary sclera (PPS), which surrounds the ONH. Pressure on the ONH and PPS

that is a continuum of the LC can cause astrocyte reactivity and compression of RGC axons that can lead to RGC death[85,86]. Genes mapped to POAG, but not IOP loci were also enriched in oligodendrocytes that form a myelin sheath around the axons of RGCs, and oligodendrocyte precursor cells found in the ON, suggesting new IOP-independent mechanisms that can affect optic nerve degeneration. The strongest enrichment for IOP was in vascular endothelial (also

**Fig. 8 | Cell type enrichment of e/sQTL-mapped genes to POAG, IOP and related trait loci in retina, optic nerve head and surrounding posterior tissues.** Significance (circle size, -log10(P-value)) and fold-enrichment (circle color) of the cell type-specificity of e/sGene mapped-GWAS locus sets for POAG cross-ancestry, POAG European (EUR) subset, IOP, physician-defined vertical-cup-to-disc ratio (VCDR) or machine learning-defined VCDR (VCDR ML) GWAS, using ECLIPSER (one-sided), shown for all cell types in retina (**a**) and optic nerve head (ONH), optic nerve (ON), peripapillary sclera, sclera, and choroid (**b**). Traits (rows) and cell types (columns) were clustered based on hierarchical clustering of Euclidean distance between cell type-specificity enrichment scores of GWAS locus sets. Red rings: experiment-wide significant (Benjamini-Hochberg (BH) FDR < 0.1). Yellow rings: tissue-wide significant (BH FDR < 0.1). Grey rings: nominal significant (P < 0.05). Colored boxes highlight trait-specific enriched cell types. **c** Cell type-specificity fold-enrichment (x-axis) in ONH and surrounding tissue cell types ranked in descending order for the POAG cross-ancestry GWAS locus set (No. loci=79). Points: fold-enrichment estimates from ECLIPSER. Error bars: 95% confidence intervals. Exact p-values (one-sided) are provided in Supplementary Data 35. **d** Differential

expression (log$_2$(Fold-change)) in astrocytes (6-Astro) in ONH compared to all other cell types in the posterior tissues is shown for the e/sGenes driving the POAG enrichment signal in astrocytes. Horizontal dashed line represents log$_2$(Fold-change) of 0.375 (FC = 1.3) and FDR < 0.1, used as the cell type-specificity enrichment cutoff. **e** Expression profile of e/sGenes driving the POAG cross-ancestry enrichment signal in astrocytes (red box) shown across all ONH and surrounding tissue cell types. Colorbar represents gene expression z-scores computed by comparing each gene's average expression in a given cell type to its average expression across all types divided by the standard deviation of all cell type expression averages. Bubble size is proportional to percentage of cells expressing the gene (log(TPK + 1) > 1). **f** LocusCompare plot of -log$_{10}$(P-value) of POAG cross-ancestry GWAS meta-analysis relative to -log$_{10}$(P-value) of *DGKG* retina eQTL, which significantly colocalized with POAG locus rs56233426 (chr3_186411027_G_A). Points are color-coded based on LD (r$^2$) relative to the eVariant with highest eCAVIAR colocalization posterior probability (CLPP = 0.93). Cell type abbreviations are described in Supplementary Data 35.

enriched for POAG genes) and fibroblast cells primarily residing in the choroid, but also in the ONH and PPS, suggesting that vascular structural abnormalities or functional dysregulation of blood flow to the optic nerve and retina may be an important contributor to POAG[87]. It is also possible that the enrichment in the ONH vascular endothelial cells is capturing causal mechanisms acting in the vascular endothelial cells in the anterior segment that were enriched for most of the same IOP genes as in the ONH. The IOP genes were also enriched in vascular smooth muscle cells (VSMC) in the PPS and sclera. These VSMC-specific genes were enriched in lipid binding and negative regulation of cell substrate adhesion processes, suggesting a role in cytoskeleton-associated cell-cell adhesion and cell-extracellular matrix adhesion. These muscle cells may be part of the LC, as LC cells isolated from human ONH were found to stain for alpha-smooth muscle actin[83,84]. Cells in the LC produce extracellular matrix proteins to support the LC structure[84], and biomechanical strain on the LC, such as from elevated IOP, is thought to be one of the causes of RGC degeneration[88,89]. Further investigation will be needed to determine whether the smooth muscle cells found in the ONH single nucleus dataset[37] reside in the LC or in other structures such as blood vessels. We observed that many of the POAG-relevant cell types, such as the Müller glia, astrocyte, fibroblast, and vascular endothelial cells, are proliferative cells, suggesting that POAG associations may be, at least in part, reflective of cycling cells and processes.

Notably, we did not find significant enrichment of cell type-specific expression of POAG or IOP genes in RGCs, whose cell death is the key characteristic of glaucoma, but rather in neuronal support cells. This highlights the importance of targeting the support cells in new therapy development. It should be noted that multiple colocalizing POAG genes are expressed in RGCs (see Supplementary Fig. 20 and Monavarfeshani et al.[37]), however they are also expressed in other retinal cell types. Hence, the potential effect of these genes on POAG via RGCs merits further investigation. Furthermore, while several studies have suggested that microglia, specialized macrophage-like cells, may affect RGC survival[90], we did not find support for a causal role of microglia or immune cells in POAG susceptibility. A less expected result was the enrichment of POAG genes in RPE cells, which was also shown in the posterior ocular cell atlas[37]. In all, our findings in retina, ONH and the surrounding tissues propose cell types and biological processes that may be viable targets for neuroprotective therapies.

A potential limitation of our GWAS-cell type enrichment method, ECLIPSER is that it only considers genes that map to genome-wide significant loci and not subthreshold associations. We thus provided further support for our cell type enrichment results using two additional methods, stratified LD score regression and MAGMA, that

analyze multiple modest associations genome-wide in additional to known GWAS loci. Furthermore, ECLIPSER primarily considers genes whose expression is specific to one or few cell types within a tissue, as the cell type specificity scoring metric was found to be successful in identifying known pathogenic cell types for a range of complex diseases and traits in a cross-tissue single nucleus expression atlas using GTEx samples[38]. We note though that genes expressed at similar levels across most or all cell types may also contribute to disease risk or trait variation and would be missed with this approach. Future analyses will be needed to investigate the role of cellular heterogeneity within each cell type in POAG development, using single-cell enrichment methods such as scDRS[91] and scPagwas[92].

In conclusion, our work has generated new insights into POAG mechanisms, which could inform the development of novel therapies targeting IOP reduction and neuroprotection. By integrating genetic regulation and single cell expression in glaucoma-relevant ocular tissues with GWAS summary statistics we have identified known and new causal genes and biological processes, proposed key ocular cell types that may be pathogenic for glaucoma, and provided evidence for the existence of hundreds of novel genetic associations of regulatory effects for glaucoma. In the future, detection of e/sQTLs in relevant eye tissues and at cellular resolution[93,94] is expected to provide a more complete picture of the causal molecular and cellular mechanisms of POAG risk and IOP variation.

## Methods

Our research complies with all relevant ethical regulations. The committees and institutions that approved the study protocols of the various datasets analyzed in this paper, including GWAS, e/sQTL, single nucleus-RNA-seq, and Hi-C studies are specified in the relevant citations provided below.

### GWAS datasets

We applied colocalization and fine-mapping analysis to 127 GWAS loci identified in the cross-ancestry POAG GWAS meta-analysis of 34,179 cases and 349,321 controls from European, African, and East Asian populations[8], 68 GWAS loci from the GWAS meta-analysis of the European subset of 16,677 POAG cases and 199,580 controls[8], and 133 LD-independent GWAS variants in 112 loci from the IOP GWAS meta-analysis of 139,555 primarily UK Biobank (European) samples[9]. The GWAS meta-analysis summary statistics, which included p-value, effect size and standard error, were obtained from the corresponding studies. Chromosome positions were lifted over from genome build 37 (hg19) to hg38. Association results on chromosome X were only available for the POAG GWAS meta-analyses (cross-ancestry and European subset).

## GTEx and EyeGEx QTL datasets

*cis*-eQTLs and *cis*-sQTLs from 49 tissues from GTEx release v8[26] and *cis*-eQTLs from peripheral retina[27] were used in this study. Summary statistics of all variant-gene e/sQTL pairs tested in each of the 50 tissues, the significant e/sGenes and e/sVariants at FDR < 0.05, and the gene expression levels and LeafCutter[95] values are available for download from the GTEx portal (https://gtexportal.org/home/datasets). The summary statistics of all variant-gene pairs tested per gene and tissue was used as input to the colocalization analysis, and the LocusZoom (http://locuszoom.sph.umich.edu)[96] and LocusCompare plots (https://github.com/boxiangliu/locuscomparer). Plots of exon and exon junction read counts were taken from the visualizations on the GTEx portal (https://gtexportal.org). GENCODE versions 26 and 25 were used for the GTEx v8 and EyeGEx studies, respectively.

## Enrichment of POAG and IOP associations among e/sQTLs using *QTLEnrich*

To test whether genome-wide significant and nominal POAG and IOP trait associations are enriched among eQTLs and sQTLs, and to assess the contribution of e/sQTLs to these traits, we applied *QTLEnrich*[24,26] to the POAG and IOP GWAS meta-analyses summary statistics, using eQTLs and sQTLs from the 49 GTEx tissues[26] and eQTLs from peripheral retina (EyeGEx[27]). *QTLEnrich* is a rank and permutation-based method that evaluates the fold-enrichment significance of trait associations among a set of e/sQTLs in a given tissue, correcting for three confounding factors: minor allele frequency (MAF), distance to the target gene's transcription start site, and local LD[24] (for more details see Supplementary Methods). Only protein-coding and lincRNA genes were considered in this analysis. Significant tissues were determined based on an Enrichment *P*-value that passed Bonferroni correction, correcting for 50 tissues and two QTL types tested ($P < 5 \times 10^{-4}$). The adjusted fold-enrichment was used to rank the significantly enriched tissues, as this statistic is not correlated with tissue sample size or number of significant e/sQTLs per tissue[24], as observed with the colocalization analysis (Supplementary Fig. 4). For the significant trait-tissue pairs, the fraction and number of e/sVariants proposed to be associated with POAG or IOP, beyond genome-wide significance, were estimated using an empirically derived, true positive rate (Adj. $\pi_1$) approach that we implemented in the latest version of *QTLEnrich* (https://github.com/segrelabgenomics/QTLEnrich), based on Storey's analytical $\pi_1$[97] and an empirical FDR method[98] (see Supplementary Methods).

## Colocalization analysis

To identify a high confidence set of genes and regulatory mechanisms (e/sQTLs) that may be mediating the functional mechanisms underlying known common variant associations with POAG and IOP, we applied two Bayesian-based colocalization methods: eCAVIAR (https://github.com/fhormoz/caviar)[41] and *enloc* (https://github.com/xqwen/fastenloc)[42]. These methods assess the probability that co-occurring GWAS and e/sQTL signals are tagging the same causal variant or haplotype, accounting for local LD and allelic heterogeneity, using slightly different fine-mapping and colocalization approaches. They are applied to GWAS and QTL summary-level statistics enabling the analysis of large, well-powered GWAS meta-analyses, for which genotype data are not available. For the *enloc* analysis, DAP-G (https://github.com/xqwen/dap/tree/master/dap_src)[42] was used to perform fine-mapping of GWAS and e/sQTL loci to estimate the posterior probabilities of each variant in each locus being the causal variant, while eCAVIAR has the fine-mapping feature built in. At most two independent causal variants per locus were assumed with eCAVIAR, while the number of independent causal variants was not limited with *enloc*. The priors used in each method are described in the associated references[41,42]. We applied the two colocalization methods to 127 POAG cross-ancestry GWAS, 68 POAG GWAS loci from the European subset

meta-analysis, and 133 independent IOP variants (112 loci) from a primarily European study (described above). Z-scores from the GWAS and GTEx e/sQTL studies, computed as the effect size (beta) divided by the standard error of the effect size for each variant, were used as input into eCAVIAR and DAP-G. For the retina eQTLs, we computed z-scores from the variant association *p*-values assuming a chi-square distribution with 1 degree of freedom.

All GWAS loci were tested for colocalization with all eQTLs and sQTLs from 49 GTEx tissues[26] and peripheral retina eQTLs[27] that had at least 5 e/sVariants (FDR < 0.05) within the GWAS locus LD interval. An LD window around each lead GWAS variant was defined as the chromosome positions on either side containing variants within $r^2 > 0.1$, determined using 1000 Genomes Project Phase 3[99] as the reference panel, and extending an additional 50 kb on either side. For the IOP and POAG EUR loci, LD was computed using only the European samples in 1000 Genomes Project, while for the cross-ancestry POAG loci, LD was computed using the European, African, and East Asian samples in 1000 Genomes. If a GWAS variant was not found in 1000 Genomes, an LD proxy variant ($r^2 > 0.8$) was searched for in GTEx, and if not found, the nearest variant was used. PLINK v1.9 (https://www.cog-genomics.org/plink/) was used to compute LD. The interval boundaries and number of variants tested are reported in Supplementary Data 7–13. eCAVIAR and *enloc* analyses were applied to all common variants (MAF > 1%) that fell within the GWAS LD intervals and were present in both the GWAS and e/sQTL studies. The effect allele of the variants in each GWAS was aligned relative to the alternative (ALT) allele that was used as the effect allele in GTEx and EyeGEx. Colocalization analysis of the retina eQTLs was only performed using eCAVIAR. GWAS-e/sQTL-tissue combinations with a colocalization posterior probability (CLPP) above 0.01 were considered significant with eCAVIAR and/or with an RCP above 0.1 were considered significant with *enloc* based on the methods' recommendations[41,42,53]. To remove potential false positives, we filtered out variant, gene, tissue, and trait combinations where the e/sVariant with a significant colocalization result had a GWAS *p*-value above $1 \times 10^{-5}$ or whose e/sQTL *p*-values was above $1 \times 10^{-4}$ and/or did not pass FDR < 0.05 (FALSE in column 'Pass_QC_QTL_FDR05_P1E04_GWAS_P1E05' in Supplementary Data 7–12). Further details on the eCAVIAR and *enloc* analyses and quality control can be found in Supplementary Methods.

## Mendelian randomization (MR)

Mendelian randomization (MR)[43] was used to provide additional genetic support for a causal relationship between colocalizing e/sQTLs and POAG and/or IOP loci. Significant e/sVariants were used as the instrumental variable (IV) in MR to facilitate causal inference[100] (See Supplementary Methods). Two-sample MR was applied to the summary statistics of the e/sQTLs (exposure) and POAG or IOP GWAS (outcome) for all significant colocalizing loci (Supplementary Data 7–12), using the *TwoSampleMR* and *MendelianRandomization* packages in R (version 4.1.2)[101]. To avoid confounding by ancestry, MR was conducted using the European ancestry subset of the POAG GWAS and the IOP GWAS, which primarily contains European individuals. MR estimates were generated by calculating the Wald ratio, i.e., the variant-outcome association beta divided by the variant-exposure association beta[102]. Where multiple variants constituted the instrument for the candidate gene, the inverse-variance weighted (IVW) method was used as the primary method for pooling variant-specific estimates[103]. Given that the IVW approach assumes no horizontal pleiotropy, methods robust to violation of the exclusion-restriction assumption were used as sensitivity analyses. The simple-median[104], weighted-median[104], MR-Egger[105], and MR-PRESSO[106] methods were applied. Horizontal pleiotropy was tested using the Egger-intercept test and the MR-PRESSO global heterogeneity test on cases with 3 or more IV variants. $P < 0.05$ indicated the presence of horizontal pleiotropy. MR associations with Benjamini-Hochberg (BH) FDR < 0.05 for

the primary IVW/Wald ratio test were considered statistically significant. In cases where horizontal pleiotropy was found based only on the MR-PRESSO global heterogeneity test, an MR PRESSO outlier-corrected *p*-value < 0.05 was considered a significant result for horizontal pleiotropy. All MR-related statistical tests were implemented in the following pipeline: https://github.com/segrelabgenomics/TwoSampleMR_pipeline.

### Integration of retina Hi-C and epigenetic data with colocalizing POAG loci and e/sQTLs

To identify retina eQTLs or GTEx e/sQTLs that colocalized with POAG GWAS loci that may be exerting their causal effect on POAG in the retina, we inspected all POAG loci in the context of chromatin loops, *cis* regulatory elements (CREs) and super-enhancers (SEs) that were previously detected in retina from 5 postmortem non-diseased human donor eyes (2 females, 3 males; age range: 65–77)[40]. The loops were calculated from Hi-C (3D chromosome conformation capture) data, and the CREs and SEs from epigenetic data, as described in Marchal et al.[40]. The lead POAG GWAS variants and their LD proxy variants ($r^2 > 0.7$), the colocalizing e/sQTLs and LD proxy variants that are also significant e/sVariants (FDR < 0.05), and the e/sQTL target genes were inspected for overlap or closest overlapping gene with the Hi-C loops, CREs, and SEs, using the closestBed command from bedtools (v2.27.1)[107]. For retina eQTLs and GTEx e/sQTLs GENCODE versions 25 and 26 were used, respectively, to overlap genes and TSS hg38 coordinates. Colocalizing e/sGenes were proposed as putative causal genes to POAG if the e/sVariant overlapped one foot of the loop and the second foot overlapped the gene body or TSS of the target gene. CRE and SE target genes were defined if the e/sVariant and gene body or TSS of the gene overlapped the same CRE or SE. The closest target genes identified using chromatin loops for the POAG cross-ancestry GWAS loci was taken from our recently published Hi-C study (Supplemental Data 4 in Marchal et al.[40]).

### Single nucleus RNA-seq datasets and differential gene expression

We analyzed gene expression values ($log(TPK + 1)$) from four single-nucleus (sn) RNA-seq data sets from the following glaucoma-relevant regions of the eye: anterior segment[36], retina[33], macula[36], and optic nerve head and surrounding posterior tissues[37]. All tissue samples were dissected from non-diseased eye globes from post-mortem donors with no record of eye disease, and were de-identified. Description of tissue dissection and processing, single-nuclei isolation, snRNA-sequencing and cell type clustering can be found in the corresponding publications[33,36,37]. The number of cells per cell types in each of the tissues can be found in Supplementary Data 35. Differential gene expression (DGE) was applied to genes expressed in at least 5% of cells in any cell type cluster in each of the datasets, and fold-change of average gene expression in each cell type compared to all other cell types in a given tissue and the associated FDR were computed. Here is a brief description of the four datasets:

### Anterior segment

Six tissues in the anterior segment, including central cornea, corneoscleral wedge (CSW), trabecular meshwork (TM), iris, ciliary body (CB), and lens, were dissected from six donors (2 females, 4 males; age range: 30-66) within 6 h from death, as described in[36]. To be able to compare across cell types between tissues in the anterior segment, the snRNA-seq data from cornea, CSW, CB, iris, and TM were pooled, downsampled to 1000 cells per type in each tissue, and reclustered yielding 34 clusters[36]. Five clusters were identified for the lens. DGE analysis between each cell type and all other cell types was performed using the regression model in MAST[108] that corrects for the proportion of genes expressed per cell.

### Retina

Retina samples from the fovea (4 mm punch), macula (6 mm punch) and/or periphery were collected from six donors (2 females, 4 males; age range: 65-84) within 6 h from death from the Utah Lions Eye Bank, flash frozen and processed as described in Liang et al.[33]. snRNA-seq data from RGCs from a few additional donors were added to the data set, given the relevance of RGCs to glaucoma, though RGCs still only comprised about 1/250 of the total data set. DGE for each cell type in the *ml_class* level used in this study was computed using the Wilcoxon rank sum test.

### Macula

Macular samples were dissected with 8 mm punches from five donors (5 males; age range: 41–77) within 4 h of death at the University of Utah[36]. For three of the samples, RGCs were enriched by staining the nuclei with NEUN antibody (Millipore Sigma, #FCMAB317PE) followed by FACS sorting[36]. DGE for each cell type compared to all other cell types was computed using the MAST method[108].

### Optic nerve head and posterior tissues

The optic nerve head, including peripapillary tissues, was dissected with 4 mm punches from 13 donors (4 females, 9 males; age range: 30-77), the optic nerve was dissected from 7 donors, peripapillary sclera from 4 donors, sclera from 3 donors, and choroid from 5 donors, within a median of 6 h from death at either the University of Utah or Massachusetts General Hospital[37]. Thirty-six cell type clusters were identified across the five tissues. DGE was computed using the MAST method[108], comparing the cells from each cell type to all other cells, excluding cells from the same cell class similar to the cell type of interest, aside for the given cell type (e.g., excluding all fibroblast cell types when computing DGE for cell type, 5-Fibro).

### Cell type-specific enrichment of genes that map to GWAS loci for a given complex trait using ECLIPSER

To identify ocular cell types that are enriched for cell type-specific expression of genes mapped to GWAS loci of POAG, IOP and related traits, we extended a method we recently developed called ECLIPSER (Enrichment of Causal Loci and Identification of Pathogenic cells in Single Cell Expression and Regulation data; https://github.com/segrelabgenomics/ECLIPSER)[38,39], to target genes of colocalizing e/sQTLs. ECLIPSER assesses whether genes mapped to a set of GWAS loci for a given complex disease or trait are enriched for cell type-specific expression compared to the cell type specificity of genes mapped to a background (null) set of GWAS loci associated with hundreds of unrelated traits. The underlying assumption of ECLIPSER is that multiple (though not necessarily all) trait-associated genes will be more highly expressed in a given pathogenic cell type compared to non-pathogenic cell types in a tissue of action, more so than unrelated traits. The analysis consisted of the following steps: (i) Mapping genes to GWAS loci. For the POAG and IOP traits, e/sQTL colocalization analysis was used to prioritize genes in GWAS loci. For the cornea-related, VCDR and negative control traits, genes were mapped to GWAS loci if they were target genes of a GTEx or retina e/sQTL that was in LD ($r^2 > 0.8$) with the GWAS locus (since colocalization analysis for these traits was beyond the scope of this paper). The genome-wide significant variants associated with the cornea traits and negative control traits were taken from Open Targets Genetics[109], and for physician and machine learning-based VCDR measures from the corresponding published GWAS meta-analyses[12,13]. (ii) Null set of GWAS loci. We compiled a null set of GWAS loci, by selecting all genome-wide significant associations for a range of complex traits in Open Targets Genetics[109] that were taken from the NHGRI-EBI GWAS catalog and UK Biobank GWAS studies. We excluded from the null set variants associated with any ocular trait. (iii) LD clumping of loci. We collapsed GWAS variants that were in LD with each other ($r^2 > 0.8$) or that shared

a mapped gene into a single locus for the set of GWAS loci for each ocular or negative control trait and for the null set, separately, to avoid inflating the cell type enrichment results due to LD[39]. (iv) Cell type specificity locus score. We scored each GWAS locus for the ocular traits and the null set as the fraction of genes mapped to the locus that demonstrated cell type specificity (defined here as fold-change > 1.3 and FDR < 0.1). Only genes expressed in at least 5% of cells in any cell type cluster were included in the analysis. (v) Assessing cell type-specificity of GWAS locus set. We estimated a cell type specificity fold-enrichment and *p*-value per trait (GWAS locus set), tissue and cell type combination, compared to the null GWAS locus set, using a Bayesian Fisher's exact test and the 95th percentile of the null locus scores for the cell type specificity cutoff. The Bayesian approach enables estimating 95% confidence intervals of the fold-enrichment, including for traits that have few or no loci that fall above the enrichment cutoff[39]. (vi) Cell type specific disease-contributing genes. Cell type-specific genes mapped to GWAS loci whose cell type specificity locus score was equal to or above the 95th percentile enrichment cutoff in significantly enriched cell types ('leading edge loci') were proposed to influence the given complex trait in the given cell type ('leading edge genes'), though it is possible that some of these genes are affecting the given trait through other cell types. Cell types with a tissue-wide Benjamini-Hochberg FDR equal to or below 0.1, correcting for multiple cell types tested within a tissue, were considered significantly enriched for genes associated with a given trait. To test the specificity of ECLIPSER, we applied the method to eight negative control traits listed in Supplementary Data 42. To assess the robustness of the cell type enrichment results with ECLIPSER, we ran two additional cell type enrichment methods of GWAS data that consider genome-wide genetic associations beyond genome-wide significant loci: stratified LD score regression[25] and MAGMA[44] (see below).

### Cell type specific heritability enrichment of disease associations using stratified LD score regression

We applied stratified LD score regression (S-LDSC)[25] (v1.0.1. https://github.com/bulik/ldsc) to the GWAS summary statistics of the POAG cross-ancestry meta-analysis, POAG European subset meta-analysis, and IOP meta-analysis, and the four single-nucleus differential gene expression datasets described above, to evaluate the contribution of genetic variation in cell type-specific genes to trait heritability. Common variants (MAF > 1%) within or near genes specifically expressed in the different cell types (fold-change > 1.1 and FDR < 0.1) in each of the four single-nucleus eye tissue datasets described above, were considered in the S-LDSC analysis. A 100 kb windows on either side of each gene was used. The European samples in 1000 Genomes Project Phase 3[99] were used as the reference panel for computing the LD scores for all three GWAS meta-analyses. Heritability enrichment per cell type was considered significant at Benjamini-Hochberg FDR below 0.1.

### MAGMA gene-association correlation with cell type gene expression

We applied the regression-based model MAGMA (v1.10, https://ctg.cncr.nl/software/magma)[44] to the POAG cross-ancestry, POAG European subset, and IOP GWAS meta-analyses and the four single-nucleus ocular expression datasets described above, which tests for association between gene association z-scores and average gene expression per cell type, controlling for average gene expression across all cell types per tissue. Gene-based association z-scores were computed for each GWAS based on the most significant variant (SNP-wise=top) within 100 kb around each gene, as described in de Leeuw et al.[110]. The European samples in 1000 Genomes Project Phase 3[99] were used as the reference panel for the POAG EUR and IOP GWAS, while all five populations (EUR, AFR, AMR, EAS, SAS) in 1000 Genomes were used for the POAG cross-ancestry GWAS. Significance was determined at Benjamini-Hochberg FDR below 0.1. We applied conditional analysis

to all pairwise combinations of nominally significant (*P* < 0.05) cell types within a given tissue to identify cell types whose trait association signals are independent of the other significant cell type[44]. A proportional significance (PS) of the conditional *P*-value of a cell type relative to its marginal *P*-value was computed for each cell type in each cell type pair. Two cell types in a given pair with PS ≥ 0.8 were considered independently associated cell types, and a pair of cell types with PS ≥ 0.5 were considered partial-joint associations. In the case where one cell type had PS ≥ 0.5 and the second cell type a conditional *P*-value ≥ 0.05, the first cell type was retained and the second cell type was considered completely dependent on the association of the first cell type. For more details see: https://fuma.ctglab.nl/tutorial#celltype.

### Gene set enrichment analysis of POAG and IOP associated genes

We used *GeneEnrich* (https://github.com/segrelabgenomics/GeneEnrich)[24] to test whether genes proposed to affect POAG risk or IOP variation cluster in specific biological processes or mouse phenotype ontologies. *GeneEnrich* assesses enrichment of a set of genes of interest in biological pathways or other types of biologically meaningful gene sets, using a hypergeometric distribution and permutation analysis. To account for biases that could arise from the set of genes expressed in a given tissue, an empirical gene set enrichment *P*-value was computed as the fraction of 1000 to 100,000 *k* randomly sampled genes (*k* = number of significant genes, e.g., colocalizing e/sGenes) from all genes expressed in the given tissue (background set) that have a hypergeometric probability equal to or higher than that of the significant list of genes. Given the high LD in the HLA region on chromosome 6 (chr6:28510120-33480577) we removed all genes in this region from the gene set enrichment analysis, unless noted otherwise.

We applied *GeneEnrich* to three groups of POAG and IOP associated genes: (i) All 228, 118, and 279 unique target genes of eQTLs and sQTLs that colocalized with POAG cross-ancestry, POAG EUR, and IOP GWAS loci, respectively. Given that the colocalizing e/sQTLs were derived from the different GTEx tissues and retina, we used all genes expressed in any of the 49 GTEx tissues and retina as the background set of genes, and did not correct for expression levels given the differences in expression levels between the tissues. (ii) Sets of POAG and IOP colocalizing genes that were enriched in specific cell types in the eye tissues based on ECLIPSER analysis (tissue-wide FDR ≤ 0.1). For the background sets of genes, we chose all genes expressed in the GTEx or retina tissue that was most relevant for the enriched cell type (e.g., Brain for Optic nerve head; full list of tissues chosen in Supplementary Data 36). Given that the expression levels in a tissue may not fully reflect the expression levels in the particular cell type, we did not correct for expression levels in the gene set enrichment analysis of the cell type-specific gene sets. (iii) Target genes of e/sQTLs (FDR < 0.05) with top ranked POAG or IOP GWAS *P*-values (*P* < 0.05) in tissues whose e/sQTLs were enriched for trait associations based on *QTLEnrich*. Given that e/sQTLs in most tissues displayed significant enrichment, a selected set of QTL/tissue-trait pairs was chosen for gene set enrichment analysis based on the tissue having a top ranked adjusted fold-enrichment and consisting of cell types that may be relevant to glaucoma pathophysiology, such as cells-cultured fibroblasts, brain, and artery (Supplementary Data 4, 5). The background sets of genes were defined as all genes expressed in the given tissue excluding the target genes of e/sQTLs with GWAS *P* < 0.05. The expression levels of the randomly sampled genes from the background set in the permutation analysis were matched on the expression levels of the significant set of genes. We applied *GeneEnrich* to over 11,000 gene sets from four databases downloaded from MSigDB (http://www.gsea-msigdb.org/gsea/msigdb/collections.jsp): Gene Ontology (GO) with three domains: biological processes, molecular function, and cellular components, Reactome, Kyoto Encyclopedia of Genes and Genomes (KEGG), and mouse phenotype ontology gene sets from the Mouse Genome Informatics (MGI). Only gene sets with 10 to 1000 genes were tested,

and only genes that were found in the given database were included in the analysis. Statistical significance was determined using a Benjamini-Hochberg FDR below 0.1 computed per database, given extensive gene set overlap between databases. Gene sets with empirical gene set enrichment below 0.05 were considered nominally significant.

### Conditional analysis of *MYOC* POAG locus

Given our finding of significant colocalization of a *PIGC* sQTL with the POAG cross-ancestry association signal in the GWAS locus rs74315329, whose lead variant is a nonsense mutation in the *MYOC* gene, we tested whether there was a secondary independent POAG signal in this locus that might colocalize with the *PIGC* sQTL. We performed association testing on all variants on chromosome 1 conditioning on rs74315329, the lead POAG GWAS variant in the locus by applying the tool COJO (https://yanglab.westlake.edu.cn/software/gcta/) to the POAG cross-ancestry GWAS meta-analysis summary statistics on chromosome 1 (using --cojo-cond). To maintain the *MYOC* lead variant that has a MAF of ~0.001 in the initial association testing we filtered out variants with MAF < 0.0001. The effective sample size of the POAG cross-ancestry GWAS was computed based on the equation: $4/[(1/N_{cases})+(1/N_{controls})]$[111], which yielded $N = 124{,}531$ for the POAG GWAS cross-ancestry meta-analysis[8]. For the variant allele frequencies required as input to COJO, we used the European, African and East Asian samples in 1000 Genomes Project Phase 3[99], as the POAG GWAS meta-analysis is comprised of these three ancestral groups. eCAVIAR and *enloc* were applied to the residual POAG statistics of the variants in the *MOYC* locus from the conditional analysis and all overlapping e/sQTLs from the GTEx tissues and retina. To remove potential false positives, we filtered out variant, gene, tissue, and trait combinations if the significantly colocalizing e/sVariant had a GWAS *p*-value above $2 \times 10^{-5}$ or an e/sQTL *p*-value above $1 \times 10^{-4}$ and/or an FDR above 0.05 (FALSE in column 'Pass_QC_QTL_FDR05_P1E04_GWAS_P2E05' in Supplementary Data 25 and 27). We used a slightly more lenient GWAS *p*-value cutoff for the conditional analysis ($P < 2 \times 10^{-5}$) compared to the original GWAS summary statistics ($P < 1 \times 10^{-5}$) given the reduced association power of conditional analysis.

### Reporting summary

Further information on research design is available in the Nature Portfolio Reporting Summary linked to this article.

## Data availability

All data supporting the findings in this manuscript are available from the links above or in Supplementary Data. The GTEx protected data are available through the database of Genotypes and Phenotypes (dbGaP) (accession no. phs000424.v8). The processed GTEx eQTL and sQTL and EyeGEx retina eQTL summary statistics are available on the GTEx portal (https://gtexportal.org/home/datasets). The snRNA-seq data for the anterior segment and macula are available in Gene Expression Omnibus (GEO) accession number GSE199013, for the optic nerve head and posterior tissues in GSE236566, and for the retina in GSE226108. The processed data of the anterior and posterior segments can be visualized in the Broad Institute's Single Cell Portal at https://singlecell.broadinstitute.org/single_cell/study/SCP1841 and https://singlecell.broadinstitute.org/single_cell/study/SCP2298. The retina Hi-C data is accessible in GEO accession number GSE202471. The GWAS summary statistics for the POAG cross-ancestry GWAS meta-analysis and European subset meta-analysis are accessible in GEO under accession numbers GCST90011770 and GCST90011766, respectively, and for IOP are available from the corresponding publication (Khawaja et al., Nature Genetics 2018)[9]. The GWAS loci for complex traits used in the ECLIPSER analysis were downloaded from Open Targets Genetics (https://genetics.opentargets.org/). The gene sets taken from MSigDB, including Gene Ontology, Reactome and KEGG, were downloaded from: http://www.gsea-msigdb.org/gsea/msigdb/collections.jsp, and

the mouse phenotype ontology gene sets were downloaded from the Mouse Genome Informatics (MGI) website (http://www.informatics.jax.org/). All the results from our analyses (e.g., colocalization, Mendelian randomization, cell type enrichment) can be found in the Supplementary Data. The colocalization and mendelian randomization results can be viewed on our https://VisionGenomics.org portal.

## Code availability

The code of all tools used for analyses in this paper are publicly available and can be found in the URLs provided in the Methods section. The pipeline we wrote to perform the MR-related statistical tests can be downloaded from: https://github.com/segrelabgenomics/TwoSampleMR_pipeline. Custom code used to generate the plots are available upon request.

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

## Acknowledgements

We thank members of the Segrè lab for valuable comments and feedback, and Xiaoquan (William) Wen for helpful discussions on the interpretation of the colocalization results. This work was funded by NIH/NEI R01 EY031424 (A.V.S., A.R.H., J.M.R., P.A.M.), NIH/NEI P30 EY014104 (J.L.W., A.V.S.), NIH/NEI R01 EY032559 (J.L.W., A.V.S., L.R.P.), NIH/NEI R01 EY022305 (J.L.W.), and the Chan Zuckerberg Initiative (CZI) Seed Network for the Human Cell Atlas awards CZF2019-002459 (J.R.S., A.V.S., T.V.Z., W.Y., A.M.) and CZF2019-002425 (Q.L., R.C.). X.J. is supported by the University of Edinburgh and University of Helsinki joint PhD studentship program in Human Genomics. V.V. is supported by an MRC University Unit Programme grant (MC_UU_00007/10) (QTL in Health and Disease). Sr.M. was supported by NIH/NIGMS grant no. T32GM144273. P.G. is supported by an NHMRC Investigator Grant (#1173390). St.M. is supported by an NHMRC Senior Research Fellowship and an NHMRC Program Grant (APP1150144). A.P.K. is supported by a UK Research and Innovation Future Leaders Fellowship, an Alcon Research Institute Young Investigator Award and a Lister Institute for Preventive Medicine Award. This research was supported by the NIHR Biomedical Research Centre at Moorfields Eye Hospital and the UCL Institute of Ophthalmology (A.P.K.). This work is also supported by The Glaucoma Foundation (NYC) and an unrestricted challenge grant from Research to Prevent Blindness (L.R.P.), and intramural Research Program of National Eye Institute (ZIAEY000546 to A.S.).

## Author contributions

A.R.H. and A.V.S. conceived and designed the study. A.R.H., W.Y., X.J., P.A.M., J.A., Y.L., Q.L., S.R., Ar.S., K.D., Sr.M and A.V.S. analyzed the data. A.R.H., J.M.R. J.W., and A.V.S. developed new statistical methods. A.M., T.V.Z., An.S., P.G., A.P.K., St.M., R.C., J.R.S., J.L.W., and IGGC contributed data for the analysis. T.V.Z., V.V., L.R.P., St.M., J.R.S and J.L.W. provided feedback on the analysis and interpretation of the results. A.R.H. and A.V.S. wrote the manuscript with input and edits from all other authors.

## Competing interests

J.R.S. was a consultant for Biogen, but none of this work benefits or benefited from that relationship. T.V.Z. is an employee of Regeneron. A.P.K. has acted as a paid consultant or lecturer to Abbvie, Aerie, Allergan, Google Health, Heidelberg Engineering, Novartis, Reichert, Santen and Thea. L.R.P. is a paid consultant to Twenty Twenty. St.M. is a co-founder of and holds stock in Seonix Pty Ltd. J.L.W. is a recent consultant for CRISPR Therapeutics and Editas. The remaining authors declare no competing interests.

## Additional information

[1]Ocular Genomics Institute, Department of Ophthalmology, Massachusetts Eye and Ear, Boston, MA, USA. [2]Department of Ophthalmology, Harvard Medical School, Boston, MA, USA. [3]Broad Institute of Harvard and MIT, Cambridge, MA, USA. [4]Department of Molecular and Cellular Biology and Center for Brain Science, Harvard University, Cambridge, MA, USA. [5]MRC Human Genetics Unit, Institute of Genetics and Cancer, The University of Edinburgh, Edinburgh, UK. [6]Centre for Genomic and Experimental Medicine, Institute of Genetics and Molecular Medicine, The University of Edinburgh, Edinburgh, UK. [7]Neurobiology, Neurodegeneration and Repair Laboratory, National Eye Institute, National Institutes of Health, Bethesda, MA, USA. [8]Department of Molecular and Human Genetics, Baylor College of Medicine, Houston, TX, USA. [9]Centre for Evidence-Based Medicine, University of Oxford, Oxford, UK. [10]Faculty of Medicine, Imperial College London, London, UK. [11]Bioinformatics and Integrative Genomics (BIG) PhD Program, Harvard Medical School, Boston, MA, USA. [12]Harvard/MIT MD-PhD Program, Harvard Medical School, Boston, MA, USA. [13]Department of Ophthalmology and Visual Sciences, Yale School of Medicine, New Haven, CT, USA. [14]Department of Ophthalmology, Icahn School of Medicine at Mount Sinai, New York, NY, USA. [15]QIMR Berghofer Medical Research Institute,

Brisbane, QLD 4029, Australia. [16]NIHR Biomedical Research Centre, Moorfields Eye Hospital NHS Foundation Trust and UCL Institute of Ophthalmology, London, UK. [62]These authors contributed equally: Wenjun Yan, John M. Rouhana, Aboozar Monovarfeshani. ✉e-mail: ayellet_segre@meei.harvard.edu

## International Glaucoma Genetics Consortium (IGGC)

Alex W. Hewitt[17,18], Alexander K. Schuster[19], Ananth C. Viswanathan[20,21,22], Andrew J. Lotery[23,24], Angela J. Cree[24], Anthony P. Khawaja ®[16], Ayellet V. Segrè ®[1,2,3] ✉, Calvin P. Pang[25], Caroline Brandl[26,27], Caroline C. W. Klaver[28,29,30,31], Caroline Hayward[5], Chiea Chuen Khor[32], Ching-Yu Cheng[33,34,35], Christopher J. Hammond[36], Cornelia van Duijn[37,38], David A. Mackey[39], Einer Stefansson[40], Eranga N. Vithana[41], Francesca Pasutto[42], Fridbert Jonansson[43], Gudmar Thorleifsson[44], Jacyline Koh[45], James F. Wilson[5,46], Jamie E. Craig[47], Janey L. Wiggs ®[1,2,3], Joëlle E. Vergroesen[28,29], John H. Fingert[48,49], Jost B. Jonas[33,50,51,52], Kári Stefánsson[44], Kathryn P. Burdon[17], Li Jia Chen[25], Louis R. Pasquale[14], Michael Kass[53], Nomdo M. Jansonius[54,55,56], Norbert Pfeiffer[32], Ozren Polašek[57], Paul J. Foster[16], Paul Mitchell[58], Pirro G. Hysi[36], Puya Gharahkhani ®[15], Robert Wojciechowski[59], Sjoerd J. Driessen[28,29], Stuart MacGregor ®[15], Stuart W. J. Tompson[60], Terri L. Young[60], Tien Y. Wong[33,34], Tin Aung[33,34,35], Unnur Thorsteinsdottir[43,44], Veronique Vitart ®[5], Victor A. de Vries[28,29], Wishal D. Ramdas[28] & Ya Xing Wang[61]

[17]Menzies Institute for Medical Research, University of Tasmania, Hobart, TAS, Australia. [18]Centre for Eye Research Australia, University of Melbourne, Melbourne, VIC, Australia. [19]Department of Ophthalmology, University Medical Center Mainz, Mainz, Germany. [20]National Institute for Health Research Biomedical Research Centre, London, UK. [21]Moorfields Eye Hospital NHS Foundation Trust, London, UK. [22]UCL Institute of Ophthalmology, London, UK. [23]University Hospital Southampton NHS Foundation Trust, London, UK. [24]Faculty of Medicine, University of Southampton, Southampton, UK. [25]Department of Ophthalmology and Visual Sciences, The Chinese University of Hong Kong, Hong Kong, Hong Kong. [26]Department of Ophthalmology, University Hospital Regensburg, Regensburg, Germany. [27]Department of Genetic Epidemiology, University of Regensburg, Regensburg, Germany. [28]Department of Ophthalmology, Erasmus Medical Center, Rotterdam, Netherlands. [29]Department of Epidemiology, Erasmus Medical Center, Rotterdam, Netherlands. [30]Department of Ophthalmology, Radboud University Medical Center, Nijmegen, Netherlands. [31]Institute for Molecular and Clinical Ophthalmology, Basel, Switzerland. [32]Division of Human Genetics, Genome Institute of Singapore, Singapore, Singapore. [33]Singapore Eye Research Institute, Singapore National Eye Centre, Singapore, Singapore. [34]Ophthalmology & Visual Sciences Academic Clinical Program, Duke-NUS Medical School, Singapore, Singapore. [35]Department of Ophthalmology, Yong Loo Lin School of Medicine, National University of Singapore, Singapore, Singapore. [36]Departments of Ophthalmology and Twin Research and Genetic Epidemiology, King's College London, London, UK. [37]Nuffield Department of Population Health, University of Oxford, Oxford, UK. [38]Department of Genetic Epidemiology, Erasmus University Medical Center, Rotterdam, The Netherlands. [39]University of Western Australia, Centre for Ophthalmology and Vision Science, Lions Eye Institute, Nedlands, WA, Australia. [40]Faculty of Medicine, University of Iceland, Reykjavik, Iceland. [41]Singapore Eye Research Institute, Singapore, Singapore. [42]Institute of Human Genetics, Universitätsklinikum Erlangen, Friedrich-Alexander-Universität Erlangen-Nürnberg, Erlangen, Germany. [43]Faculty of Medicine, School of Health Sciences, University of Iceland, Reykjavik, Iceland. [44]deCODE genetics/Amgen Inc., Reykjavik, Iceland. [45]Singapore National Eye Centre, Singapore, Singapore. [46]Centre for Global Health Research, Usher Institute, University of Edinburgh, Edinburgh, UK. [47]Department of Ophthalmology, Flinders University, Flinders Medical Centre, Adelaide, SA, Australia. [48]Department of Ophthalmology, Carver College of Medicine, Iowa, IA, USA. [49]Institute for Vision Research, University of Iowa, Iowa, IA, USA. [50]Institute of Molecular and Clinical Ophthalmology Basel, Basel, Switzerland. [51]Department of Ophthalmology, Medical Faculty Mannheim, Heidelberg University, Heidelberg, Germany. [52]Privatpraxis Prof Jonas und Dr Panda-Jonas, Heidelberg, Germany. [53]Washington University, St Louis, MO, USA. [54]Faculty of Medical Sciences, University of Groningen, Groningen, Netherlands. [55]Department of Ophthalmology, University of Groningen, Groningen, Netherlands. [56]University Medical Center Groningen, Groningen, Netherlands. [57]School of Medicine, University of Split, Split, Croatia. [58]Centre for Vision Research, Department of Ophthalmology and Westmead Institute for Medical Research, University of Sydney, Sydney, NSW, Australia. [59]Johns Hopkins Bloomberg School of Public Health, Baltimore, MD, USA. [60]Department of Ophthalmology and Visual Sciences, University of Wisconsin, Madison, WI, USA. [61]Beijing Institute of Ophthalmology, Beijing Tongren Hospital, Capital Medical University, Beijing Ophthalmology and Visual Sciences Key Laboratory, Beijing, China.

