## [Peer Review File · Nature Communications]

Integrating genetic regulation and single-cell expression with GWAS prioritizes causal genes and cell types for glaucomaReviewers' comments:

Reviewer #1 (Remarks to the Author):

In the manuscript by Hamel, et al, the Authors provide rigorous co-localization analysis of GWAS loci implicated in primary open angle glaucoma (POAG) and the regulation of intraocular pressure with eQTL and sQTL loci. These studies are performed to better understand the molecular etiology of POAG driven by individual genomic loci, thereby assigning singular genes to GWAS associations. In addition, the Authors use these newly identified genes in combination with single-nucleus RNA-sequencing to examine the relevant tissues/cell types that contribute to disease pathogenesis. Altogether, the paper presents a heroic effort, including a mountain of data and implicating numerous novel genes/loci/cell types in the regulation of POAG and IOP.

The amount of information packaged into this one paper, however, also makes the paper very hard to follow. The Authors make a point to address specific GWAS/e/sQTL loci and the proposed affected genes (appreciated), but the implications of these novel findings gets lost within the long, run-on sentences filled with acronyms and percentages. Additionally, as a reader having to constantly reference the wealth of data (>30 supplemental tables and 17 supplemental figures), I was frequently having to re-read multiple sections of the manuscript to figure out the important findings in each section. A revised manuscript should streamline the text.

An additional concern of this manuscript is the source of the snRNA-seq data. Some of the data seems to have been previously published, but other data seems to have been generated in the course of this study. The methods sections includes details of cell capture/preparation/etc. However, none of the standard sn/scRNA-seq figures are presented within the manuscript, making it difficult to assess the quality of the snRNA-seq data. As little of this data is presented, it is difficult to assess the utility of the 'cell type specificity' analyses. I was also unable to find websites/accession numbers for any of the referenced snRNA-seq data.

Reviewer #2 (Remarks to the Author):

Hamel et al present the integrative analysis of combining e/sQTLs, single-cell expression data from the glaucoma-relevant eye tissues and pathway information with primary open-angle glaucoma (POAG) and intraocular pressure (IOP) genetic associations to dissect potential casual genes, pathways and cell types associated with POAG risk. The study provides statistical support of colocalization analysis for proposing causal genes for 60% of the IOP and POAG risk loci. Furthermore, by using scRNA-seq data of glaucoma-

relevant eye tissues, the authors uncovered the colocalizing genes were enriched in known and less well-characterized cell types, such as fibroblasts in aqueous outflow pathways, vascular cells in the anterior segment, astrocytes and Muller glia in retina and optic nerve head. I found the majority of the study well-grounded from a methodological point of view, and conclusions well-supported through statistical and computational evidence. It provides valuable insights in identifying glaucoma-critical genes and cell types through the integrative analyses of GWAS, e/sQTL, and scRNA-seq data. There were specific major or minor questions should be addressed.

1.The authors only used two colocalization methods (e.g., eCAVIAR and enloc) for fine-mapping the risk genes associated with glaucoma. There was no evaluation of the causal effects of these colocalizing genes for glaucoma. How the authors infer the causal effects? What is the definition of causal genes in the manuscript?

2.If sought to infer the causal relationships, the method of Mendelian randomization is a good alternative approach for assessing the causal effects of genes on diseases by using GWAS summary statistics and eQTL.

3.Current investigation primarily focused on their analyses for the colocalizing genes across multiple tissues. How many colocalizing genes identified from glaucoma-associated tissues (e.g., retina eQTLs or other proxies).

4.The authors used their own developed software ECLIPSER to identify POAG-associated cell types based on 228, 118, and 279 e/sGenes colocalized with POAG CA, POAG EUR, and IOP GWAS loci. Multiple significant finding were identified, such as fibroblasts, vascular cells. As I known, two methods of LDSC-SEG, MAGMA have been extensively used for searching disease-associated cell types, did you assess those two methods for identifying disease-related cell types? What differences in results were there between ECLIPSER and LDSC-SEG/ MAGMA?

5.Whether the cellular composition of single cell RNA-seq data influence the ECLIPSER results? The authors should give some supports for this point. For example, if a dataset consists mostly of a specific cell type of astrocyte, this will impact the detection of enrichment in astrocyte or not?

6.Significant enrichment was found in astrocyte and Muller glia for POAG genes and IOP genes. Are POAG and IOP both associated with the same genes that are specifically expressed in the same cell type, or both traits are associated with different sets of genes that are both specific to the same cell type?

7.The conditioning analyses for these cell type enrichment analyses should be provided. If conditioning on the overlapped genes with IOP, did the enrichment in astrocyte and Muller glia (or other significant cell types) for POAG remain significant?

Reviewer #3 (Remarks to the Author):

The manuscript focuses on prioritizing potential causal variants to identify genes, regulatory mechanisms, pathways, and cell types that contribute to POAG risk. First, applying the QTL/GWAS

colocalization methods (CAVIAR and enloc) on the e/sQTL datasets from 49 GTEx tissues and retina, they prioritized potential causal genes for about 60% of POAG and IOP-associated variants (that were previously reported). This strategy led them to propose a single gene to about 80 POAG/IOP-associated loci (most of which were protein-coding genes). Next, they applied gene set enrichment analysis to determine biological or physiological processes enriched for the proposed causal genes. Amongst the pathways identified are extracellular matrix organization, cell adhesion, and vascular development. Additionally, they employed an ECLIPSER test to determine whether the proposed genes are enriched for cell type-specific expression (using a combination of e/sQTL and scRNA-seq datasets) and that allowed identification of specific cell types through which the proposed genes mediate its effect. They found significant enrichment of POAG and IOP genes in cell types, including fibroblasts (potentially involved in conventional and unconventional aqueous outflow pathways), astrocytes and Müller glia in the retina and optic nerve head.

This is an initial attempt that employs a comprehensive, high throughput approach for a large-scale prioritization of causal variants associated with IOP and POAG. Overall, the methods used and data analysis are appropriate. However, there are a number of limitations, importantly a lack of depth along functional lines.

- 1) The study is largely dependent on publicly available or previously published gene expression-based datasets (e/sQTL, RNA-seq) to prioritize variants associated with POAG/IOP. The study lacks any functional or mechanistic depth.
- 2) They have made use of e/sQTL datasets available through GTEx and previously published eQTL datasets of peripheral retina. Ocular tissues are not represented in GTEx, which is a major limitation, and thus, deriving any conclusions could be misleading.
- 3) The study points to an alternative splice donor site in exon 4 of TMCO1 mRNA identified in cultured fibroblasts from GTEx that result in a longer exon 4, which is associated with decreased POAG risk. A thorough characterization of the splice variants of TMCO1 in trabecular meshwork (TM) cells could have helped extend this finding to a more biologically relevant cell type.
- 4) There are fundamental flaws with using cell-type enrichment of a given gene as a strategy to identify cell-types mediating the effect of potential causal variants. It is too simplistic in scope. This is further compounded by the use of ECLIPSER, a method that considers only those genes whose expression is specific to one or few cell types within a tissue, and not those that are more broadly expressed. Each cell type provides a unique context (for example, with respect to its genomic landscape or interacting partners). Just because a gene is enriched in a specific cell type does not mean it is likely the causal gene.
- 5) Cell types such as Muller cells and astrocytes are highly represented in cell-type enrichment of proposed genes. The glial cell types are highly plastic and are likely to be triggered by slight disturbances. Enzymatic dissociation of tissue is a step that is integral to single-cell RNA seq analysis. Such treatments could alter gene expression patterns, especially in a cell type like glia, which could result in inherent biases. Moreover, enrichment in immune cell types, such as lymphocytes could result from similar biases. The cell-type enrichment data could have been strengthened by localization studies in intact tissues (in situ/ immunohistochemistry) for at least some of the top hits.

Reviewers' comments:

Reviewer #1 (Remarks to the Author):

In the manuscript by Hamel, et al, the Authors provide rigorous co-localization analysis of GWAS loci implicated in primary open angle glaucoma (POAG) and the regulation of intraocular pressure with eQTL and sQTL loci. These studies are performed to better understand the molecular etiology of POAG driven by individual genomic loci, thereby assigning singular genes to GWAS associations. In addition, the Authors use these newly identified genes in combination with single-nucleus RNA-sequencing to examine the relevant tissues/cell types that contribute to disease pathogenesis. Altogether, the paper presents a heroic effort, including a mountain of data and implicating numerous novel genes/loci/cell types in the regulation of POAG and IOP.

The amount of information packaged into this one paper, however, also makes the paper very hard to follow. The Authors make a point to address specific GWAS/e/sQTL loci and the proposed affected genes (appreciated), but the implications of these novel findings gets lost within the long, run-on sentences filled with acronyms and percentages. Additionally, as a reader having to constantly reference the wealth of data (>30 supplemental tables and 17 supplemental figures), I was frequently having to re-read multiple sections of the manuscript to

figure out the important findings in each section. A revised manuscript should streamline the text.

Response: We thank the reviewer for raising this important point. We agree that the manuscript in its original form was densely written making it difficult for the reader to distill the key findings. We have shortened the main text and put many details in the Supplementary Note, thereby making the Results section more readable. We have also highlighted the main novel findings and conclusions in the Results and Discussion.

The main findings are:

- Identifying a high confidence set of eQTLs and sQTLs and target genes that may contribute to POAG risk and/or IOP variation.
Identifying regulatory mechanisms and genes that are likely to affect POAG in the retina, independent of IOP.
Identifying biological processes implicated by genetics in POAG pathogenicity, including extracellular matrix organization, cell adhesion, vascular development, and retina morphology.
Identifying novel or less well-established pathogenic cell types for POAG, such as astrocytes and Müller glia cells in the retina, fibroblasts in the peripapillary sclera that surrounds the optic nerve head, and oligodendrocytes in the optic nerve head and optic nerve, in addition to previously implicated cell types in POAG pathogenesis, such as fibroblasts and vascular cells in the aqueous humor outflow pathways that influence IOP levels, by integrating GWAS loci, e/sQTLs, and single cell data. Our cell type enrichment results suggest that targeting neuronal support cells may be important in preventing retinal ganglion cell death.

An additional concern of this manuscript is the source of the snRNA-seq data. Some of the data seems to have been previously published, but other data seems to have been generated in the course of this study. The methods sections includes details of cell capture/preparation/etc. However, none of the standard sn/scRNA-seq figures are presented within the manuscript, making it difficult to assess the quality of the snRNA-seq data. As little of this data is presented, it is difficult to assess the utility of the 'cell type specificity' analyses. I was also unable to find websites/accession numbers for any of the referenced snRNA-seq data.

Response: We thank the reviewer for pointing this out. Indeed, all the single nucleus (sn) RNA-seq data analyzed in this paper were not published when we submitted our manuscript. However, the snRNA-seq data of 6 tissues in the anterior segment and of the macula were published in van Zyl *et al.* PNAS 2022, while our manuscript was under review, and the retina snRNA-seq atlas from Rui Chen's lab was recently published in Liang *et al.*, Cell Genomics 2023. The single nucleus atlas of the optic nerve head (ONH) and adjacent posterior tissues from Josh Sanes' lab was posted on bioRxiv (Monavarfeshani*, Yan* *et al.*, 2023) and is currently in press at PNAS. These papers or preprints demonstrate the high quality and careful curation of the cell types identified with appropriate single cell clustering, normalization and histochemistry staining in the different eye tissues. We added references to these papers or preprint in our revised manuscript (References #33, 36, 37), and moved much of the details of the single nucleus dissociation, sequencing experiments and analyses to the Supplementary Note. We also added the accession numbers for the four snRNA-seq datasets in a section called 'Data availability' that we added at the end of the manuscript (Page 28, lines 923-927). Of note, since the review of our manuscript, the optic nerve head (ONH) single nucleus dataset was enhanced by Josh Sanes' lab to include additional posterior tissue structures adjacent to

the ONH, including the optic nerve, peripapillary sclera, peripheral sclera and choroid. We have thus updated our cell type enrichment analysis of e/sGenes that colocalized with POAG and IOP GWAS loci using this expanded ONH and posterior tissue single-nucleus dataset. While the key results are the same, the expanded posterior tissue cell atlas provided higher resolution of the cell types in the ONH and adjacent tissues implicated by our work in glaucoma, compared to the previous snRNA-seq dataset version.

Reviewer #2 (Remarks to the Author):

Hamel et al present the integrative analysis of combining e/sQTLs, single-cell expression data from the glaucoma-relevant eye tissues and pathway information with primary open-angle glaucoma (POAG) and intraocular pressure (IOP) genetic associations to dissect potential causal genes, pathways and cell types associated with POAG risk. The study provides statistical support of colocalization analysis for proposing causal genes for 60% of the IOP and POAG risk loci. Furthermore, by using scRNA-seq data of glaucoma-relevant eye tissues, the authors uncovered the colocalizing genes were enriched in known and less well-characterized cell types, such as fibroblasts in aqueous outflow pathways, vascular cells in the anterior segment, astrocytes and Muller glia in retina and optic nerve head. I found the majority of the study well-grounded from a methodological point of view, and conclusions well-supported through statistical and computational evidence. It provides valuable insights in identifying glaucoma-critical genes and cell types through the integrative analyses of GWAS, e/sQTL, and scRNA-seq data. There were specific major or minor questions should be addressed.

1. The authors only used two colocalization methods (e.g., eCAVIAR and enloc) for fine-mapping the risk genes associated with glaucoma. There was no evaluation of the causal effects of these colocalizing genes for glaucoma. How the authors infer the causal effects? What is the definition of causal genes in the manuscript?

Response: We define a causal gene as a gene for which there is statistical support that it might contribute to disease risk or trait variation, in the case of our study through changes in expression levels or alternative splicing. We acknowledge that experimental validation in relevant cellular contexts is needed to conclude with high confidence that a gene prioritized with e/sQTL-GWAS colocalization analysis is causal to a disease, and thus made sure to use the words 'proposed', 'putative', 'potential', or 'prioritized' before the words 'causal gene' when mentioned throughout the manuscript (e.g. Page 3, line 80). We had also previously noted in the Discussion (Page 15, lines 498-500) that causal effects of the proposed genes on POAG or IOP need to be experimentally followed up. We clarified the definition of 'causal genes' in the manuscript in the Results on page 4, lines 122-125: "Given the widespread e/sQTL enrichment of POAG and IOP associations, we used the e/sQTLs in all 49 GTEx tissues and retina eQTLs to propose putative causal genes that may underlie genome-wide significant loci for these traits."

In regards to the comment that we applied only two colocalization methods, most studies, such as Kim-Hellmuth *et al.*, Nature Comm 2017, Nathan *et al.*, Nature Genetics 2022, and D'Antonio *et al.*, Nature Comm 2023, apply a single colocalization method, and often use the coloc method that has a simplistic assumption of a single causal variant underlying the GWAS and e/sQTL loci. In this study, we applied two methods with slightly different fine-mapping and colocalization approaches to increase our discovery power of colocalizing e/sQTLs and GWAS loci, and to identify a high confidence set of colocalizing loci that are significant with both methods. Furthermore, the two methods we applied, eCAVIAR and enloc allow for the assumption of

multiple independent causal variants (allelic heterogeneity) per locus, which is often the case for GWAS and e/sQTL loci.

The reviewer raises a good point about the limitation of colocalization analysis in inferring a causal relationship between e/sQTLs and GWAS loci. While colocalization analysis assesses the probability that co-occurring association signals, such as GWAS and e/sQTL signals, share a causal variant or haplotype, we cannot rule out the possibility that the causal effect on POAG or IOP may be through another mechanism other than change in gene expression or alternative splicing (horizontal pleiotropy). We had previously acknowledged this limitation in the Discussion, and now expanded on this point in the Supplementary Note (Page 6, lines 181189). To further address the reviewer's comment about evaluating the causal effect of the colocalizing genes on glaucoma, we performed two-sample Mendelian Randomization (MR) analysis on all of the significantly colocalizing e/sQTL and GWAS loci pairs found with eCAVIAR and/or enloc (see more details in our response below to comment #2 of Reviewer #2). This led to the identification of a high confidence set of genes (75% of colocalizing e/sGenes) that may contribute to POAG risk and/or IOP variation.

2. If sought to infer the causal relationships, the method of Mendelian randomization is a good alternative approach for assessing the causal effects of genes on diseases by using GWAS summary statistics and eQTL.

Response: We agree with the reviewer that Mendelian randomization (MR) is a good method to add support for a causal relationship between the colocalizing e/sQTLs and GWAS signals, as we had previously noted in the Discussion. We have now applied two sample Mendelian randomization analysis to all the significantly colocalizing e/sQTLs in the 49 GTEx tissues and retina and GWAS loci based on eCAVIAR and/or enloc, to identify a high confidence set of causal genes and regulatory mechanisms for POAG and IOP. We used the Wald ratio or inverse-variance weighted (IVW) method as the primary analysis, and performed sensitivity analyses using four additional statistical tests. We also tested for Horizontal pleiotropy using the Egger-intercept test and MR-PRESSO global heterogeneity test. 3711 significant QTL type-gene-tissue combinations acting on 348 (75%) of the colocalizing e/sGenes had significant MR results for POAG and/or IOP GWAS at $FDR < 0.05$, which passed the horizontal pleiotropy tests. The MR results are summarized in Supplementary Table 29, and the analysis is described in the Results section on page 8, lines 248-265, the Methods section on pages 21-22, lines 702721, and the Supplementary Note (Pages 10-11, lines 310-343). We also added a note about the MR analysis in the Discussion on Page 14, lines 463-465. We have updated Table 1 to reflect a high confidence list of putative causal genes for POAG and/or IOP based on both significant colocalization and MR results.

3. Current investigation primarily focused on their analyses for the colocalizing genes across multiple tissues. How many colocalizing genes identified from glaucoma-associated tissues (e.g., retina eQTLs or other proxies).

Response: We thank the reviewer for raising this very relevant question. While retina is the only ocular tissue for which eQTLs is currently available, multiple GTEx tissues contain cell types that are proxies for pathogenic cell types for glaucoma, such as brain, which contains neurons and neuronal support cells, or arterial tissue that contains vascular cells. We had previously included in the colocalization results table (Supplementary Table 13, previously Supplementary Table 8) and in Table 1 a column that specifies whether the colocalizing eGenes are based on retina eQTLs (column AH in Supplementary Table 13, and 'retina eQTL' column in Table 1). We now also added a sentence in the Results section (Page 5, lines 138-139) that notes the

number of POAG and/or loci that colocalized with a retina eQTL: “Eighteen retina eQTLs colocalized with 13 POAG and/or IOP loci (Column AH in Supplementary Table 13).”

To provide additional support to the proposed regulatory effects on POAG in retina based on colocalization and Mendelian randomization analyses, we integrated the colocalizing e/sQTL-POAG loci with Hi-C (3D chromatin capture loops) data, and cis-regulatory elements (CREs) and super-enhancers (SEs) inferred from epigenetic data from healthy human retina, generated recently in the lab of Anand Swaroop (Marchal *et al.*, Nature Comm 2022). The results are summarized in Supplementary Table 22. We highlighted several vignettes (*SLC2A12*, *RERE* and *CDKN2B-AS1* POAG loci) in Figure 5, where retinal Hi-C and epigenetic data strongly support a causal effect of e/sQTLs on POAG in retina. The analysis is described in the Results (Pages 6-7, lines 199-216) and Methods (Page 22, lines 722-738) sections, and discussed in the Discussion on page 16, lines 512-516.

Per the reviewer’s comment on highlighting colocalizing e/sQTLs in proxy tissues, we added a sentence in the Results section (Pages 4-5, lines 135-138) that points out the tissues with the highest number of colocalizing genes, which also contain cell types relevant to POAG pathogenicity, including nerve (retinal ganglion cells and macroglial cells), adipose (lipid-related processes), artery (Schlemm’s canal), and fibroblasts (outflow pathway). The full list of tissues with colocalizing e/sQTLs is given in Supplementary Tables 7-13. Since we observed that the number of significantly colocalizing e/sGenes with POAG or IOP GWAS loci from each GTEx tissue highly correlated ($R^2=0.72$) with tissue sample size, which is also associated with number of detected e/sQTLs per tissue (Page 5, lines 139-142; see also Supplementary Fig. 4), we did not put too much emphasis on the tissues in which e/sQTLs colocalized with POAG and/or IOP loci for downstream analyses, as noted on Page 5, lines 139-144.

Of note, to identify GTEx tissues whose e/sQTLs are enriched for POAG or IOP associations, while correcting for the potential confounding effect of tissue sample size on the relative contribution of e/sQTLs on POAG and IOP, we applied *QTLEnrch* (Gamazon*, Segre* *et al.*, Nat Genet 2018, GTEx consortium, Science 2020), a method we previously developed and applied to a range of complex traits (GTEx consortium, Science 2020). *QTLEnrch* tests for enrichment of GWAS associations amongst eQTLs or sQTLs in a given tissue, while adjusting for tissue sample size, e/sQTL number, distance to TSS, MAF, and local LD. Based on this analysis we found significant enrichment of hundreds of POAG or IOP associations (genome-wide significant and nominal) among eQTLs or sQTLs in a range of GTEx tissues. The top ranked tissues contained POAG relevant cell types including artery, fibroblasts, nerve, and brain tissues. This analysis is described in the Results section on page 4, lines 103-120, and demonstrates the relevance of using e/sQTLs from non-ocular GTEx tissues to prioritize causal mechanisms and genes underlying POAG and IOP GWAS loci. Additional details can be found in the Supplementary Note on page 2, lines 39-67, and pages 6-8, lines 196-239.

4. The authors used their own developed software ECLIPSER to identify POAG-associated cell types based on 228, 118, and 279 e/sGenes colocalized with POAG CA, POAG EUR, and IOP GWAS loci. Multiple significant findings were identified, such as fibroblasts, vascular cells. As I know, two methods of LDSC-SEG, MAGMA have been extensively used for searching disease-associated cell types, did you assess those two methods for identifying disease-related cell types? What differences in results were there between ECLIPSER and LDSC-SEG/MAGMA?

Response: We thank the reviewer for raising this important point and helping assure that the conclusions of our paper are robust. We applied the two additional methods proposed by the

reviewer, stratified-LD score regression (LDSC-SEG) (Supplementary Table 48) and MAGMA (Supplementary Table 49), to all of our ocular single-nucleus RNA-seq datasets and to the POAG cross-ancestry GWAS meta-analysis, POAG European subset meta-analysis, and IOP GWAS meta-analysis summary statistics. We compared the cell type enrichment results from these two methods to those obtained with ECLIPSER (Supplementary Figure 22 and Supplementary Table 50). We observed relatively high concordance in the results (Average Pearson's correlation coefficient $r=0.53$, range: 0.18-0.86; Supplementary Table 50). The primary cell types found with ECLIPSER, including ciliary and trabecular meshwork fibroblasts and vascular cells in the anterior segment, optic nerve head fibroblasts, and retinal macroglial cells, were also found to be significant with S-LDSC and MAGMA (Supplementary Figure 22 and Supplementary Tables 48 and 49), thus providing supportive evidence for the top enriched cell types reported in the paper. These results are summarized in the Results section (Pages 12-13, lines 398-409), Methods (Page 25, lines 829-839 for S-LDSC and pages 25-26, lines 841-859), Supplementary Tables 48-50, and displayed in Supplementary Figure 22. We also added a sentence in the Discussion (Page 18, lines 603-606) on the added support from these methods for the proposed pathogenic cell types for POAG and/or IOP.

There were some significant cell types found with ECLIPSER that were not significant with LDSC and/or MAGMA and vice versa, which might be expected, as ECLIPSER only considers genes in genome-wide significant GWAS loci, while S-LDSC and MAGMA consider also more modest associations genome-wide. The main difference we observed was with the enrichment of POAG loci in oligodendrocytes in the optic nerve head that was found only with ECLIPSER, and not with S-LDSC or MAGMA. This might suggest that the enrichment in oligodendrocytes is primarily driven by genes in the strongest genetic effect loci. We have added a sentence pointing this out on page 12, lines 405-407.

5. Whether the cellular composition of single cell RNA-seq data influence the ECLIPSER results? The authors should give some supports for this point. For example, if a dataset consists mostly of a specific cell type of astrocyte, this will impact the detection of enrichment in astrocyte or not?

Response: This is an excellent question. To address this point, we tested whether cellular abundance in our four single nucleus datasets was correlated with the ECLIPSER cell type fold-enrichment or enrichment p-values. We did not find significant correlation of number of cells per cell type with the cell type enrichment results in any of the single nucleus RNA-seq datasets for POAG, IOP, VCDR, VCDR_ML_Alipanahi, central corneal thickness or corneal hysteresis GWAS (Pearson $R^2 < 0.2$, $P > 0.12$). The Pearson correlation coefficient results are summarized in Supplementary Table 47, and the cell counts per cell type were added to the ECLIPSER results table in Supplementary Table 38. We added a sentence on this analysis in the Results section on page 12, lines 396-397 and in the Supplementary Note on Page 5, lines 166-169.

6. Significant enrichment was found in astrocyte and Muller glia for POAG genes and IOP genes. Are POAG and IOP both associated with the same genes that are specifically expressed in the same cell type, or both traits are associated with different sets of genes that are both specific to the same cell type?

Response: Thank you for raising this informative question. The POAG-colocalizing genes were strongly enriched in astrocytes ($P=0.00087$, $FDR=0.013$) and Müller glia cells ($P=0.005$, $FDR=0.037$) in the retina, while the IOP genes were only nominally enriched in retinal astrocytes ($P=0.032$), but not in Müller glia cells ($P>0.25$) (Supplementary Table 38). We thus added a comparison of the POAG and IOP genes driving the enrichment signals in astrocytes in

our single-nucleus retina datasets. We found that only one third of the 12 POAG genes driving astrocyte enrichment were common with IOP. We also ran ECLIPSER on POAG-only or IOP-only loci (also addressed below in point #7 of Reviewer #2). We found that the astrocyte and Müller glia cell enrichment are significant for POAG-only loci or shared loci, but not for IOP-only loci (Supplementary Table 41 and Supplementary Fig. 17), suggesting that the POAG enrichment in retinal astrocytes and Müller glia cells is largely independent of IOP. We added a sentence on this analysis in the Results (Page 11, lines 358-360), and additional details in the Supplementary Note (Page 5, lines 136-147), as follows:

“We assessed the extent to which the enrichment of POAG genes in retinal astrocytes was driven by IOP-independent associations. Only one third of the 12 POAG genes driving astrocyte enrichment in retina were common with IOP (*DGKG, FMNL2, GAS7, LPP*) (Supplementary Fig. 16d, f). Furthermore, when applying ECLIPSER to POAG-only or IOP-only loci or shared loci, significant enrichment (FDR<0.08) was found in astrocytes and Müller glia cells in retina and macula solely for POAG-only and shared loci, but not for IOP-only loci (Supplementary Table 41 and Supplementary Fig. 17), suggesting an IOP-independent effect of astrocytes and Müller glia cells on glaucoma.”

7. The conditioning analyses for these cell type enrichment analyses should be provided. If conditioning on the overlapped genes with IOP, did the enrichment in astrocyte and Müller glia (or other significant cell types) for POAG remain significant?

Response: We thank the reviewer for raising this very useful point and suggestion. To address this question we performed the following two analyses:

(1) We assessed the cell type enrichment in the anterior ocular segment, retina, and optic nerve head and surrounding posterior tissues, using ECLIPSER, considering the following three subsets of genes: (i) genes that map only to POAG loci, (ii) genes that map only to IOP loci, and (iii) genes that map to both POAG and IOP GWAS loci. The cell type enrichment results for the POAG and IOP common and independent loci, and the genes driving the significant enrichment are summarized in Supplementary Table 41 and Supplementary Figures 15, 17, and 19, and described in the Results section on page 11, lines 342-348, page 11, 358-360, and page 12, lines 387-389.

In summary, we found that the retinal astrocytes and Müller glia cells were significantly enriched for POAG-only loci or POAG/IOP shared loci (FDR<0.08), but not for IOP-only loci (Supplementary Table 41 and Supplementary Fig. 17), suggesting an IOP-independent effect of astrocytes and Müller glia cells on glaucoma risk. In the optic nerve head, we found that the enrichment in oligodendrocytes and oligodendrocyte precursor cells (OPCs) was specific to POAG-only loci, and enrichment in vascular endothelium and mural cells was specific to IOP-only loci, while the enrichment in fibroblasts in the posterior tissues was shared with POAG and IOP (Supplementary Table 41 and Supplementary Fig. 19). In the anterior segment, the enrichment in ciliary and trabecular meshwork fibroblasts in the outflow pathways was driven by shared IOP and POAG loci, while enrichment in pericytes in was driven by IOP-only loci. We did not find significant enrichment for POAG-only loci in the anterior segment cell types, supporting IOP-dependent mechanisms in the anterior segment affecting POAG risk, as expected (Supplementary Table 41 and Supplementary Fig. 15).

(2) To test whether the POAG and IOP associations driving the enrichment signals for the significant cell types in each tissue are dependent or independent of each other, we performed conditional analysis between the different cell types per tissue using MAGMA's multivariate

regression model. This analysis compares two models: the association of POAG or IOP gene association z-scores as a function of average gene expression per cell type, with and without controlling for the average gene expression in a second cell type (e.g, comparing trait association with astrocytes relative to Muller glia cells). The results for all pairwise comparisons of all significant ($P < 0.05$) cell types in the anterior segment, retina and optic nerve head and surrounding tissues are summarized in Supplementary Table 51. This analysis revealed that the enrichment in astrocytes was independent of the enrichment in Müller Glia, but not vice versa, suggesting that astrocytes may play a more important role in glaucoma pathogenicity than Müller Glia cells. Furthermore, we found that the enrichment in the fibroblasts in the ciliary muscle that is part of the unconventional outflow pathway is independent of the enrichment in fibroblasts in the trabecular meshwork that makes up the conventional outflow pathway. We describe these results in the Result section on Page 13, lines 409-417, and in the Methods on page 26, lines 851-859.

We also added a sentence in the Results section that compares the overlap between the POAG genes driving the enrichment in astrocytes and those driving the enrichment in Müller Glia cells (Page 11, lines 354-357):

“A quarter (*YAP1*, *LPP*, *TRIB2*) of the 12 POAG cross-ancestry genes driving the astrocyte enrichment were common with Müller glia cells (Supplementary Fig. 16d, e), suggesting both shared and distinct processes between the two cell types.”

Reviewer #3 (Remarks to the Author):

The manuscript focuses on prioritizing potential causal variants to identify genes, regulatory mechanisms, pathways, and cell types that contribute to POAG risk. First, applying the QTL/GWAS colocalization methods (CAVIAR and enloc) on the e/sQTL datasets from 49 GTEx tissues and retina, they prioritized potential causal genes for about 60% of POAG and IOP-associated variants (that were previously reported). This strategy led them to propose a single gene to about 80 POAG/IOP-associated loci (most of which were protein-coding genes). Next, they applied gene set enrichment analysis to determine biological or physiological processes enriched for the proposed causal genes. Amongst the pathways identified are extracellular matrix organization, cell adhesion, and vascular development. Additionally, they employed an ECLIPSER test to determine whether the proposed genes are enriched for cell type-specific expression (using a combination of e/sQTL and scRNA-seq datasets) and that allowed identification of specific cell types through which the proposed genes mediate its effect. They found significant enrichment of POAG and IOP genes in cell types, including fibroblasts (potentially involved in conventional and unconventional aqueous outflow pathways), astrocytes and Müller glia in the retina and optic nerve head.

This is an initial attempt that employs a comprehensive, high throughput approach for a large-scale prioritization of causal variants associated with IOP and POAG. Overall, the methods used and data analysis are appropriate. However, there are a number of limitations, importantly a lack of depth along functional lines.

1) The study is largely dependent on publicly available or previously published gene expression-based datasets (e/sQTL, RNA-seq) to prioritize variants associated with POAG/IOP. The study lacks any functional or mechanistic depth.

Response: Our comprehensive work proposes multiple hypotheses with mechanistic insights for functional follow up, many of which are novel. Although experimentally testing these

hypotheses is beyond the scope of this paper, to address the reviewer's comment about lack of mechanistic depth, we have integrated genome topology (Hi-C) and epigenetic data from human retina with all the significantly colocalizing e/sQTLs and POAG GWAS loci to provide additional functional support for a subset of e/sQTLs and genes proposed to affect POAG in the retina, based on colocalization and Mendelian randomization analyses. This analysis identified several strong candidate genes that may affect POAG independent of IOP. We describe these results on Pages 6-7, lines 199-216 and highlight three examples in Figure 5. The details of the analysis are in the Methods section on Page 22, lines 721-737, and we discuss an example in the Discussion on Page 16, lines 512-516.

While the e/sQTL data of the GTEx tissues and retina are publicly available data, all the single nucleus (sn) RNA-seq data analyzed in this paper were not published or publicly available when we first submitted our manuscript, and the analysis of the optic nerve head was novel. During the revisions of our paper, the snRNA-seq atlases of the ocular anterior segment and retina were published in van Zyl *et al.*, PNAS 2022 and Liang *et al.* Cell Genomics 2023, respectively. The single nucleus atlas of the optic nerve head and surrounding posterior tissues generated in the Sanes lab was recently posted on bioRxiv (Monavarfeshani*, Yan* *et al.*, 2023) and is currently in press at PNAS. We have updated the references to these papers in our revised manuscript (References #33, 36, 37).

Of note, since the review of our manuscript, the single nucleus atlas of the optic nerve head was expanded by the Sanes lab to include additional posterior tissues adjacent to the optic nerve head, including the optic nerve, peripapillary sclera, peripheral sclera and choroid. We have thus updated our cell type enrichment analyses of e/sGenes colocalizing with POAG and IOP GWAS loci using this enhanced posterior tissue single-nucleus dataset. Even though the key results are the same, the expanded cell atlas provided a higher resolution of the glaucoma implicated cell types in the optic nerve head and adjacent tissues compared to the previous single nucleus dataset version.

2) They have made use of e/sQTL datasets available through GTEx and previously published eQTL datasets of peripheral retina. Ocular tissues are not represented in GTEx, which is a major limitation, and thus, deriving any conclusions could be misleading.

Response: We agree that the lack of e/sQTLs from other glaucoma relevant tissues, aside for retina, is a limitation of this study, and we acknowledge this limitation in the Introduction (Page 3, lines 84-86): “Furthermore, genetic regulatory effects in relevant ocular tissues are limited, reported to date only in retinal tissues²⁷⁻³⁰ and have not yet been detected at cellular resolution

in other parts of the eye.” And discuss this point in the Discussion:

Page 16, lines 525-526: “There are several reasons why we may not have found colocalizing e/sQTLs for 40% of the loci. First, some of the causal genes or regulatory effects may be specific to regions in the eye or rare cell types for which we do not yet have representative e/sQTLs.”

Page 19, lines 618-620: “In the future, detection of e/sQTLs in relevant eye tissues and at the cellular level^{90, 91} is expected to provide a more complete picture of the causal molecular and cellular mechanisms of POAG risk and IOP variation.”

We are in fact working on generating a GTEx resource for several glaucoma-relevant tissues, including the outflow pathways and the optic nerve head, but this is work in progress.

Despite this, e/sQTLs in non-ocular tissues may be relevant for glaucoma pathogenicity, partly because many of the GTEx tissues consist of cell types that may be pathogenic to glaucoma,

such as brain, artery, and fibroblasts, and partly because it has been shown that there is abundant sharing of e/sQTL across multiple tissues (GTEx consortium, Science 2020). We demonstrate the relevance of GTEx e/sQTLs for POAG in our paper, by testing whether GTEx e/sQTLs are enriched for multiple POAG or IOP associations, while correcting for potential confounders using QTLEnrch (Results: Page 4, lines 103-120 and Supplementary Note: Page 2, lines 39-67). We found significant enrichment of POAG and IOP associations among e/sQTLs in multiple GTEx tissues. The strongest enrichment was found in brain, cells cultured fibroblasts, esophagus, and artery. These tissues share cell types that may be pathogenic to glaucoma such as astrocytes in brain that may be a proxy for retinal astrocytes, vascular cells in artery that may be a proxy for the lymphatic vessels in the aqueous outflow pathways, and cultured fibroblasts that may share genetic programs with the fibroblasts in the ocular outflow pathways. We now moved this section in our manuscript that describes this enrichment analysis from the end to the beginning of the Results section (Page 4, lines 103-120) to support the use of e/sQTLs in non-ocular tissues in proposing potential causal genes and regulatory mechanisms for POAG and IOP GWAS loci. We also discuss this point in the Discussion on Page 14, lines 451-457.

We further emphasize in the revised manuscript the colocalization results found with retina eQTLs (Page 5, lines 138-139): “Eighteen retina eQTLs colocalized with 13 POAG and/or IOP loci (Column AH in Supplementary Table 13).”, and highlight an example that is retina-specific (Page 12, lines 375-377): “*DGKG*, diacylglycerol kinase gamma, whose retina-specific eQTL colocalized (CLPP=0.96) with POAG cross-ancestry association (Fig. 7f) displayed the strongest cell type specificity in ONH (Fig. 7d) and retinal astrocytes (Supplementary Fig. 16d).” Furthermore, we added an analysis that integrates Hi-C loops and epigenetic data from human retina with colocalizing e/sQTLs with POAG loci to provide additional functional support for e/sQTLs that may affect POAG risk in retina (Figure 5, Pages 6-7, lines 199-216), as mentioned in our response to point #1 of Reviewer #3.

3) The study points to an alternative splice donor site in exon 4 of *TMCO1* mRNA identified in cultured fibroblasts from GTEx that result in a longer exon 4, which is associated with decreased POAG risk. A thorough characterization of the splice variants of *TMCO1* in trabecular meshwork (TM) cells could have helped extend this finding to a more biologically relevant cell type.

Response: We agree with the reviewer that inspecting the splice event in *TMCO1* in relevant eye cells will be of interest to understand how altered isoform expression of this gene may be affecting POAG risk. However, inspecting the splicing event in *TMCO1* in the trabecular meshwork (TM) fibroblast cells is beyond the scope of this paper, and it is not fully clear in what ocular cell type/s *TMCO1* is exerting its causal effect on POAG. To add further functional characterization of *TMCO1*, we added a plot of the expression profile of *TMCO1* in our single-nucleus RNA-seq datasets from the anterior and posterior ocular tissues shown in Supplementary Figure 7. We found that *TMCO1* is expressed in multiple ocular cell types and tissues, in addition to the trabecular meshwork cells, and thus it is not clear yet through what cell type/s *TMCO1* is affecting POAG and IOP. We added the following sentence in the Results on Page 6, lines 182-185: “*TMCO1* is expressed in different cell types in the anterior and posterior parts of the eye, including lymphatic and fibroblast cells in the conventional and unconventional outflow pathways, vascular and immune cells in the anterior and posterior segments, and macroglial cells in the retina (Supplementary Figure 7).”

4) There are fundamental flaws with using cell-type enrichment of a given gene as a strategy to identify cell-types mediating the effect of potential causal variants. It is too simplistic in scope.

This is further compounded by the use of ECLIPSER, a method that considers only those genes whose expression is specific to one or few cell types within a tissue, and not those that are more broadly expressed. Each cell type provides a unique context (for example, with respect to its genomic landscape or interacting partners). Just because a gene is enriched in a specific cell type does not mean it is likely the causal gene.

Response: Thank you for raising this important point and giving us the opportunity to clarify the utility of ECLIPSER. We realized that we failed to adequately explain and present the strong support for the use and value of ECLIPSER in identifying relevant pathogenic cell types for a range of complex traits using single-nucleus expression data from GTEx samples, which we published in Eraslan *et al.*, *Science*, 2022 (Fig. 6 and Fig. S27-S31). ECLIPSER is based on the assumption that many (though not all) disease-causing genes in GWAS loci will be more highly expressed in a pathogenic cell type compared to non-pathogenic cell types in a relevant tissue, more so than genes that affect unrelated diseases manifested in other tissues. Testing various cell type scoring metrics for GWAS loci on single-nucleus RNA-seq data from 8 GTEx tissues and 21 complex diseases and traits whose pathology is manifested in these tissues and for which at least one pathogenic cell type is known, we found that the cell type-specificity scoring metric performed the best with respect to distinguishing cell types that are more likely to be pathogenic to a given complex trait over other cell types (Eraslan *et al.*, *Science* 2022 - PMID: 35549429), compared to other scoring metrics, such as average gene expression levels per GWAS locus. We now added a reference to this work in the Introduction on Page 3, lines 91-93: “Using a method we recently developed ECLIPSER^{38,39}, we show that cell type-specific enrichment of genes mapped to GWAS loci of complex diseases and traits can help identify cell types of action for diseases in relevant tissues^{38,39}.” We also added a sentence in the Methods section (page 24, lines 789-792) that justifies the approach used by ECLIPSER.

We agree with the reviewer that “Just because a gene is enriched in a specific cell type does not mean it is likely the causal gene.” The cell type specific enrichment suggests that many of the genes in the GWAS loci driving the enrichment signal are exerting their causal effect through the enriched cell type, though we agree that some of the genes may be affecting the trait through a different cell type. We now clarified this point in the Methods section (Page 25, lines 816-819). We also added a sentence in the Discussion on Page 18, lines 606-611 that notes that there may be disease-causing genes that are expressed across many or all cell types in a given tissue, and that those would be missed in our analysis, since the primary goal of ECLIPSER was to identify the pathogenic cell types.

To further address the reviewer’s concern about the utility of ECLIPSER, we applied two additional cell type enrichment methods: stratified-LD score regression (S-LDSC) and MAGMA (for more details see our response to comment #4 of Reviewer 2), which provide support for ECLIPSER as a strategy to identify pathogenic cell types for complex diseases and traits. MAMGA, a regression-based model, assesses the association of all gene association z-scores with the genes’ average expression levels per cell type, controlling for average expression across all cell types. S-LDSC assesses enrichment of complex trait heritability among variants within or around genes that are specifically expressed in a given cell type. These methods yielded highly overlapping results with ECLIPSER (Average Pearson’s correlation coefficient $r=0.53$, range: 0.18-0.86; Supplementary Table 50). The primary cell types found with ECLIPSER were also found to be significant with S-LDSC and MAGMA, including ciliary and TM fibroblasts and vascular cells in the anterior segment, retinal macroglial cells, and fibroblasts in the peripapillary sclera and optic nerve head (Supplementary Figure 22 and Supplementary Tables 48 and 49). We describe these analyses in the Results on Pages 12-13, lines 398-409 and display the results in Supplementary Figure 22.

To demonstrate the specificity of ECLIPSER, we applied the method to eight negative control (ocular and non-ocular) traits that are unrelated to POAG and all our single nucleus RNA-seq datasets (see our response to comment #5 below).

5) Cell types such as Muller cells and astrocytes are highly represented in cell-type enrichment of proposed genes. The glial cell types are highly plastic and are likely to be triggered by slight disturbances. Enzymatic dissociation of tissue is a step that is integral to single-cell RNA seq analysis. Such treatments could alter gene expression patterns, especially in a cell type like glia, which could result in inherent biases. Moreover, enrichment in immune cell types, such as lymphocytes could result from similar biases. The cell-type enrichment data could have been strengthened by localization studies in intact tissues (in situ/ immunohistochemistry) for at least some of the top hits.

Response: We thank the reviewer for raising this important point. We understand the concern about the enrichment in glial cells being an artifact due to the effect of the tissue dissociation protocol. Recent studies have shown that transcriptional reactivity to cellular dissociation is mostly found in single cell dissociation experiments, and not as much in single nucleus dissociation experiments (see Fig. 2F and Supplementary Figure S14A, B in Eraslan *et al.*, Science 2022, PMID: 35549429). In this study they scored a published dissociation gene expression signature from van den Brink *et al.*, Nature Methods 2017 (doi:10.1038/nmeth.4437) in snRNA-seq and scRNAseq data from GTEx lung samples, which included immune cells, fibroblasts, epithelial cells, and endothelial cells. Expression of tissue dissociation signatures was accentuated in scRNA-seq, but not snRNA-seq. Nevertheless, to address this concern, we ran ECLIPSER on eight negative control traits that are unrelated to POAG, both ocular (age-related macular degeneration (AMD), macular thickness, cataract, eye color) and non-ocular (asthma, atopic dermatitis, melanoma, lung cancer), and on all our single nucleus RNA-seq datasets: anterior segment, retina, and optic nerve head and posterior tissues. The results are summarized in Supplementary Table 46 and Supplementary figure 21. The negative control traits showed enrichment in relevant cell types, such as pigmented cells for eye color, lens in the anterior segment for cataract, retinal pigment epithelium (RPE) in the macula and vascular cells in the optic nerve head for AMD, immune cells for asthma and atopic dermatitis, and melanocytes for melanoma (Supplementary Table 46, Fig. 6a and Supplementary Fig. 21). Of note, no significant enrichment was found in astrocytes or Müller Glia cells in retina for any of the negative control traits, which suggests that the enrichment of POAG genes in macroglial cells in retina was specific to glaucoma. In macula, we observed nominal enrichment of AMD genes and significant enrichment of atopic dermatitis in astrocytes in the macula, which may be capturing some true signal for these traits. Of note, we did not find enrichment of POAG genes in immune related cells, including microglia and macrophages, which as mentioned by the reviewer may also be affected by single cell dissociation protocols. We did though find enrichment of autoimmune diseases, such as asthma and atopic dermatitis, in immune cells.

We added a sentence on this negative control analysis in the Results section on Page 12, lines 393-395, pointing to more details in the Supplementary Note (Page 5, lines 152-169), where we discuss the concern of the effect of nuclei dissociation on gene expression in glial cells.

Given this raised concern, we also tested whether cell abundance per tissue may be affecting the ECLIPSER cell type enrichment results. We did not find significant correlation of the ECLIPSER enrichment p-value and number of cells per cell type for any of the tissues or GWAS tested (Pearson correlation of determination $R^2 < 0.2$, $P > 0.12$). Results are summarized in the

Supplementary Table 47, and noted in the Results on Page 12, lines 396-397, and in the Supplementary Note on Page 5, lines 166-169.

Finally, while histochemical staining is beyond the scope of this paper, to strengthen our cell type enrichment results, we ran two additional cell type enrichment methods: stratified-LD score regression (S-LDSC) and MAGMA, as described in our response to the reviewer 3's comment #4 above. MAGMA and S-LDSC supported the key cell types found with ECLIPSER, including astrocytes and Müller Glia cells in retina.

REVIEWERS' COMMENTS

Reviewer #1 (Remarks to the Author):

The Authors have addressed my reviews, including providing the source data for the scRNA-seq datasets and attempting to streamline the text.

Since all the sc/snRNA-seq data is now published, any reference to data acquisition no longer is needed in the Methods sections.

Most of the 'disease-relevant tissues' or cell type assignments have signatures of proliferative cells, including Muller glia and astrocytes which are transcriptionally similar to progenitor populations, vascular tissue, or cell types that are themselves proliferative. Is it possible that the gene associations are more reflective of cycling cells than

While the Authors have addressed these reviews, the biological significance of correlative findings remains unaddressed.

Reviewer #2 (Remarks to the Author):

I appreciate the authors that have done a lot of work for answering my concerns. In general, they have addressed my questions. Due to it has used quite long time revision, two new methods have been proposed for distinguishing the association of disease and cell populations at single-cell resolution (scPagwas published in Cell Genomics, 2023, and scDRS published in Nature Genetics, 2022). Previous methods, such as LDSC-SEG, MAGMA, ECLIPSER, RolyPoly, have been employed to incorporate GWAS and scRNA-seq data to identify predefined cell types associated with complex traits. However, these approaches largely overlook the considerable heterogeneity of individual cells within cell types.

In the current study, the authors provide multiple evidence to support the significant enrichment in astrocyte and Muller glia for POAG and IOP. As we know, there exist heterogeneity within the identified cell types. It can not be all cells in the cell types associated with POAG and IOP. I know the authors have performed a plenty of analyses in the study. I still think the authors should cite these two methods and provide some sentences to elucidate this putative heterogeneity within cell type in their discussion section.

Reviewer #3 (Remarks to the Author):

The manuscript is an attempt at prioritizing causal variants to identify genes, regulatory mechanisms, pathways, and cell types that contribute to POAG risk. Most of the concerns have been carefully addressed by including additional analysis and supportive data.

Additional functional support for the e/sQTLs was provided by integrating genome topology (Hi-C) and epigenetic data from the human retina.

Although there is a lack of e/sQTLs from other glaucoma-relevant tissues, aside from the retina, the authors have now included enrichment analysis to support/justify the use of e/sQTLs in non-ocular tissues. Similarly, the authors have provided justification for the use of fibroblasts to characterize TMCO1 splice variants.

The justification for the use of ECLIPSER has now been well substantiated and two additional cell type enrichment methods, namely stratified-LD score regression (S-LDSC) and MAGMA were employed to support the use of ECLIPSER to identify pathogenic cell type.

The authors have provided justification for tissue dissociation-related artifacts by clarifying the value of snRNA-seq and have also assessed negative control traits to rule out changes in gene expression in Muller glia and immune cells as being artifacts.

Overall, the manuscript is much improved and brings in a lot of value.

REVIEWERS' COMMENTS

Reviewer #1 (Remarks to the Author):

The Authors have addressed my reviews, including providing the source data for the scRNA-seq datasets and attempting to streamline the text.

Since all the sc/snRNA-seq data is now published, any reference to data acquisition no longer is needed in the Methods sections.

We removed the sentences in the Methods section that describe the snRNA-seq data acquisition and instead point the reader to the relevant references (see page 23, lines 779-781). We have also removed Supplementary Tables 35-37 that provide the donor and eye tissue sample information (e.g. sample ID, age, sex, ancestry) for the snRNA-seq datasets analyzed in the manuscript, as these tables are now available in the corresponding publications referenced in our paper.

Most of the 'disease-relevant tissues' or cell type assignments have signatures of proliferative cells, including Muller glia and astrocytes which are transcriptionally similar to progenitor populations, vascular tissue, or cell types that are themselves proliferative. Is it possible that the gene associations are more reflective of cycling cells than

While the Authors have addressed these reviews, the biological significance of correlative findings remains unaddressed.

We thank the reviewer for this insightful point. We added a sentence that notes this point in the Discussion section on Page 18, lines 607-609: "We observed that many of the POAG-relevant cell types, such as the Müller glia, astrocyte, fibroblast, and vascular endothelial cells, are proliferative cells, suggesting that POAG associations may be, at least in part, reflective of cycling cells and processes."

Reviewer #2 (Remarks to the Author):

I appreciate the authors that have done a lot of work for answering my concerns. In general, they have addressed my questions. Due to it has used quite long time revision, two new methods have been proposed for distinguishing the association of disease and cell populations at single-cell resolution (scPagwas published in Cell Genomics, 2023, and scDRS published in Nature Genetics, 2022). Previous methods, such as LDSC-SEG, MAGMA, ECLIPSER, RolyPoly, have been employed to incorporate GWAS and scRNA-seq data to identify predefined cell types associated with complex traits. However, these approaches largely overlook the considerable heterogeneity of individual cells within cell types.

In the current study, the authors provide multiple evidence to support the significant enrichment in astrocyte and Muller glia for POAG and IOP. As we know, there exist heterogeneity within the identified cell types. It can not be all cells in the cell types associated with POAG and IOP. I know the authors have performed a plenty of analyses in the study. I still think the authors should cite these two methods and provide some sentences to elucidate this putative heterogeneity within cell type in their discussion section.

We thank the reviewer for pointing out the new methods that account for cellular heterogeneity in testing for enrichment of GWAS associations within a given cell type. We have added a sentence in the Discussion on Page 19, lines 631-633 that notes this point: "Future analyses will be needed to investigate the role of cellular heterogeneity within each cell type in POAG development, using single-cell enrichment methods, such as scDRS⁹¹ and scPagwas⁹²."

Reviewer #3 (Remarks to the Author):

The manuscript is an attempt at prioritizing causal variants to identify genes, regulatory mechanisms, pathways, and cell types that contribute to POAG risk. Most of the concerns have been carefully addressed by including additional analysis and supportive data.

Additional functional support for the e/sQTLs was provided by integrating genome topology (Hi-C) and epigenetic data from the human retina.

Although there is a lack of e/sQTLs from other glaucoma-relevant tissues, aside from the retina, the authors have now included enrichment analysis to support/justify the use of e/sQTLs in non-ocular tissues. Similarly, the authors have provided justification for the use of fibroblasts to characterize TMCO1 splice variants.

The justification for the use of ECLIPSER has now been well substantiated and two additional cell type enrichment methods, namely stratified-LD score regression (S-LDSC) and MAGMA were employed to support the use of ECLIPSER to identify pathogenic cell type.

The authors have provided justification for tissue dissociation-related artifacts by clarifying the value of snRNA-seq and have also assessed negative control traits to rule

out changes in gene expression in Muller glia and immune cells as being artifacts.

Overall, the manuscript is much improved and brings in a lot of value.

We thank the reviewer for their valuable comments and suggestions in the first round of reviews, which greatly helped strengthen the conclusions of our manuscript.